# Increased matrix stiffness promotes fibrogenesis of hepatic stellate cells through AP-1-induced chromatin priming
Wenxue Zhao[1,6], Weihong Yuan[2,6], Tian Dong[2,6], Wei Qi[3], Zhijie Feng [3] ✉, Cheng Li [1] ✉ & Yujie Sun [4,5] ✉

Matrix stiffness has significant effects on cell behavior, however, less is known regarding the epigenomic and transcriptional regulation underling the effect of matrix stiffness on cells. In this study, we use an in vitro system to assess the phenotypic shifts of hepatic stellate cells (HSCs) following changes in matrix stiffness, and integrate multi-omics with imaging and biochemical assays to investigate the molecular mechanisms. We show that cells cultured on a stiff matrix display more accessible chromatin sites, which consist of primed chromatin regions that become more accessible prior to the upregulation of nearby genes. These regions are enriched in fibrosis-associated genes that function in cytoskeletal organization and response to mechanical stimulus. We also identify activation of p-JUN in response to the stiff matrix and promoting phenotypic shifts. The identified chromatin accessibility-dependent effect of matrix stiffness may be responsible for various fibrotic diseases and provide insight into intervening approaches.

Cells are subjected to a wide variety of mechanical stimulus, acting in multiple biological processes or diseases. For example, matrix stiffness, which refers to the rigidity of the extracellular matrix (ECM) surrounding cells, exerts various effects on cellular behavior such as proliferation, differentiation, migration, and ECM production[1]. During development, mechanical cues can guide the differentiation of stem cells[2]. Additionally, in most solid tumors, matrix stiffness alters the force-generation capability and metastatic potential of the cancer cells[3]. While it is known that matrix stiffness can induce changes in cytoskeletal organization and the expression of adhesion molecules, the manner by which matrix stiffness influences the expression and activity of epigenetic regulators and transcription factors, which play critical roles in the regulation of gene expression and cell fate, remains unclear. Moreover, fibrotic diseases, ECM dysregulation, wound healing, and tumorigenesis are intimately associated with alterations in the mechanical microenvironment[4–7]. Matrix stiffness has been implicated in fibrotic progression; however, the exact mechanisms underlying its contribution to disease pathogenesis remain elusive.

Recent advances have shown that mechanical force can regulate chromatin organization and that mechanosensitive transcription factors participate in the regulation of gene expression and cell state[8,9]. Considering the diverse phenotypic shifts that occur in hepatic stellate cells (HSCs) during the development and regression of liver fibrosis, the epigenetic changes induced by different matrix stiffness may be important contributors to shifts in cellular state[10]. The epigenome can act as a bridge connecting genotype and phenotype, and many studies have revealed that certain epigenomic changes can precede or foreshadow changes in gene expression[11,12]. For example, transient nuclear deformation and cyclic tensile stretch alter the levels and distribution of histone H3 trimethylation at lysine 9 and histone H3 trimethylation at lysine 27 (H3K27me3) to promote cell reprogramming or maintain genome integrity[13,14]. In addition, many transcription factors, such as YAP/TAZ and MRTF, are regulated by the mechanical stimuli[15,16]; however, the relationship between chromatin state and mechanotransduction during these processes is unclear, and the manner by which gene expression is regulated by chromatin remodeling needs to be explored further. Moreover, a recent advancement in multi-omics sequencing technologies, which enables joint profiling of the transcriptome and chromatin conformation, uncovered that widespread chromatin interactions are rewired before transcriptional activation[17].

Here, we established an in vitro system to explore the function of matrix stiffness on cell phenotypic shifts. LX-2, a human HSC line, was

[1]School of Life Sciences, Center for Bioinformatics, Center for Statistical Science, Peking University, Beijing, China. [2]School of Life Sciences, Peking University, Beijing, China. [3]Department of Gastroenterology, The Second Hospital of Hebei Medical University, Hebei Key Laboratory of Gastroenterology, Hebei Institute of Gastroenterology, Hebei Clinical Research Center for Digestive Diseases, Shijiazhuang, Hebei, China. [4]Biomedical Pioneering Innovation Center (BIOPIC), Peking University, Beijing, China. [5]National Biomedical Imaging Center, College of Future Technology, Peking University, Beijing, China. [6]These authors contributed equally: Wenxue Zhao, Weihong Yuan, Tian Dong. ✉e-mail: 26300056@hebmu.edu.cn; cheng_li@pku.edu.cn; sun_yujie@pku.edu.cn

seeding on hydrogel, the stiffness of which was adjusted to physiologically relevant conditions of cirrhotic and normal liver tissues. The myofibroblastic differentiation of HSC is a critical event in liver fibrosis and is part of the final common pathway to cirrhosis in chronic liver disease, and the matrix is reported a determinant of its differentiation[18]. During the progression of fibrosis, changes in biomechanical factors are mainly reflected in the reorganization and stiffening of the ECM. The continuous accumulation of ECM eventually results in an increase in tissue stiffness. And crucially, HSCs have the ability to sense these changes in stiffness, which then triggers their activation and prompts them to secrete even more ECM, thereby creating a vicious cycle[19]. Our focus on studying cell responses to stiffness in the context of fibrogenesis is precisely because stiffness acts as a key trigger in this vicious cycle. To investigate the mechanism underlying the regulation of HSC phenotypic shifts by matrix stiffness, we performed RNA-seq and ATAC-seq at two timepoints using cells exposed to an artificial matrix with two different stiffnesses, which allowed us to study the sequence of chromatin state changes and transcriptional regulation. Firstly, we found that cells cultured on a stiff matrix display a myofibroblastic phenotype. Meanwhile, chromatin accessibility and gene expression undergo reprogramming according to different matrix stiffnesses. Integrated analysis of the sequencing data identified primed chromatin sites that become accessible prior to the onset of gene expression, where the genes adjacent to these sites are mechanoresponsive and involved in fibrosis. Furthermore, we reveal that AP-1 family transcription factors play a key role in the reconstruction of chromatin accessibility and gene regulation in response to a stiff matrix. In summary, our data provide key insights into the molecular mechanism underlying the activation of HSCs by a stiff matrix and further explore the manner by which gene expression is reprogrammed through chromatin priming. These results also provide an explanation of the function of mechanical signaling in cell behavior, epigenomic landscaping, and transcription, which has important implications for better understanding biomechanics and disease pathogenesis.

## Results

### Matrix stiffness induces shifts in the fibrotic phenotype and transcriptome

To decode the mechanism underlying the induction of cell phenotypic shifts by mechanical signaling, we utilized a well-studied in vitro system in which matrix stiffness can be tuned to mimic that of physiological tissue[4], allowing us to study the effect of mechanical stimulus on cell behavior. Human LX-2 HSCs, a commonly used cell line for the study of liver disease[20–22], were seeding on hydrogels possessing a Young's moduli similar to that of normal (~2 kPa, soft) or cirrhotic (~40 kPa, stiff) liver tissue[23,24] for four days. When the HSCs cultured on tissue culture plastic for 7 days (similar to LX-2 cells in our work) were then transferred to either a soft or stiff matrix for further cultivation, the cells took a few days transiting from a rounded shape to a more spread, myofibroblast-like phenotype[25]. These results indicated that quickly after being transferred from TCP to matrices, cells start to set on activation and eventually adapt to the new mechanical environment. To assess their baseline state before being subjected to the hydrogel matrices and the subsequent changes, we measured the expression changes of active HSCs marker genes by qPCR of LX-2 cells at several critical timing points, including cells on TCP, resuspension, reseeding, and culturing on matrices for several days (Fig. 1a, b and Supplementary Fig. 1a). The results showed that when cells are transferred from TCP to a matrix, they firstly undergo a reversibility process with the gene expression patterns reset to quiescent HSC state with in several hours to one day, which is likely in accordance to the process of cells being trypsin digestion, resuspension, reseeded on the matrices, and re-adherence. Subsequently, the cells were observed to gradually establish a new steady state on either a stiff or soft matrix during the next 2–3 days. Based on these results, we collected cells cultured for 2 days and 4 days on matrices to study their mechanoresponsive mechanisms (Fig. 1c).

Firstly, we characterized the phenotype of cells on different hydrogels at both time points. Immunofluorescence of α-smooth muscle actin (α-

SMA) stress fibers was used to identify activated HSCs since it is a common marker of fibrotic HSCs. The cells cultured on the stiff matrix displayed well-organized actin stress fibers than those on the soft matrix (Fig. 1d and Supplementary Fig. 1b). To quantitatively detect the cytoskeletal gene, the expression of *ACTA2* was slightly up-regulated after 2 days of culture on a stiff matrix, and significantly up-regulated after 4 days of culture (Fig. 1b). These results demonstrate that LX-2 cells cultured on a stiff matrix display a fibrotic phenotype consistent with that of activated primary HSCs during fibrosis[18]. Next, we asked the question whether the shift toward a myofibroblastic phenotype is accompanied by global transcriptomic changes. RNA-seq uncovered genome-wide changes in the transcriptional landscape in response to mechanical stimulus after two and four days. The expression of *VCAN* (Versican), markers of fibrotic cells[24], was increased in cells cultured on the stiff matrix. The expression levels of *MMP2* and *MMP9*, encoding matrix metalloproteinases that degrade the ECM, were decreased during fibrosis (Fig. 1e, f). A greater number of genes were differentially expressed at four days than at two days, and this alteration in gene expression is consistent with those reported in previous studies on active HSCs in liver fibrosis[26–28]. Gene Ontology (GO) enrichment analysis shows that the upregulated genes are enriched in fibrotic pathways and biological processes associated with activated HSCs (Supplementary Fig. 1c, d). Next, we examined the implications of the soft versus stiff matrices for recapitulation of the in vivo quiescent and active HSCs. By comparing with the gene expression of HSCs from cirrhotic patients vs. healthy donors, cells on the soft and stiff matrices were respectively enriched with the signatures of quiescent and active HSCs, indicating their potential in respectively stimulating the gene expression patterns of in vivo quiescent or active HSCs (Supplementary Fig. 1e, f). We performed the secretory proteomic approach using liquid chromatography-tandem mass spectrometry (LC-MS/MS) to analyze the ECM deposition of cells on different matrices over periods of 2 and 4 days. The results revealed that the expression of core ECM proteins in cells cultured on the stiff matrix was generally upregulated compared to those cultured on soft substrates (Fig. 1g, h). The core ECM proteins, including collagens, ECM glycoproteins, and proteoglycans, assemble and remodel ECM and regulate cellular functions. Compositions of ECM are qualitatively and quantitatively dynamic during tissue fibrosis[29]. This finding is consistent with previous research, in which Wu et al., using LC-MS/MS, analyzed and found that the expression of core ECM proteins in the fibrotic livers of CCl₄-treated mice was also upregulated[30]. Taken together, these results show that a stiff matrix induces the transition toward a fibrotic phenotype, in addition to the upregulation of fibrosis-related genes in HSCs. The stiff matrix-induced fibrotic phenotype became increasingly pronounced during continuous stimulation.

### Increased matrix stiffness is accompanied by dynamic chromatin accessibility

To explore the mechanisms underlying transcriptomic changes in response to a stiff matrix, we assessed the chromatin accessibility dynamics using transposase-accessible chromatin with sequencing (ATAC-seq). Analysis of differentially accessible peaks (DAPs) on day 4 revealed 3786 significantly more accessible peaks in cells cultured on the stiff matrix, with only 256 peaks found to be significantly less accessible (Fig. 2a). The genes adjacent to the more accessible regions in cells cultured on the stiff matrix for four days were enriched in cell proliferation and adhesion pathways (Supplementary Fig. 2a). To reveal the changes that occur during the HSC activation process and further study the function of alterations in chromatin state in gene regulation, ATAC-seq was performed using cells cultured on the stiff matrix for two days. A total of 750 more accessible peaks and 147 less accessible peaks were found (Fig. 2a). The genes adjacent to the more accessible regions in cells cultured on the stiff matrix for two days were enriched in protein phosphorylation and signal transduction pathways as well as wound healing, which is known to be associated with the activated HSC phenotype (Supplementary Fig. 2b). These data demonstrated that culturing HSCs in a stiff matrix induced a greater number of more accessible

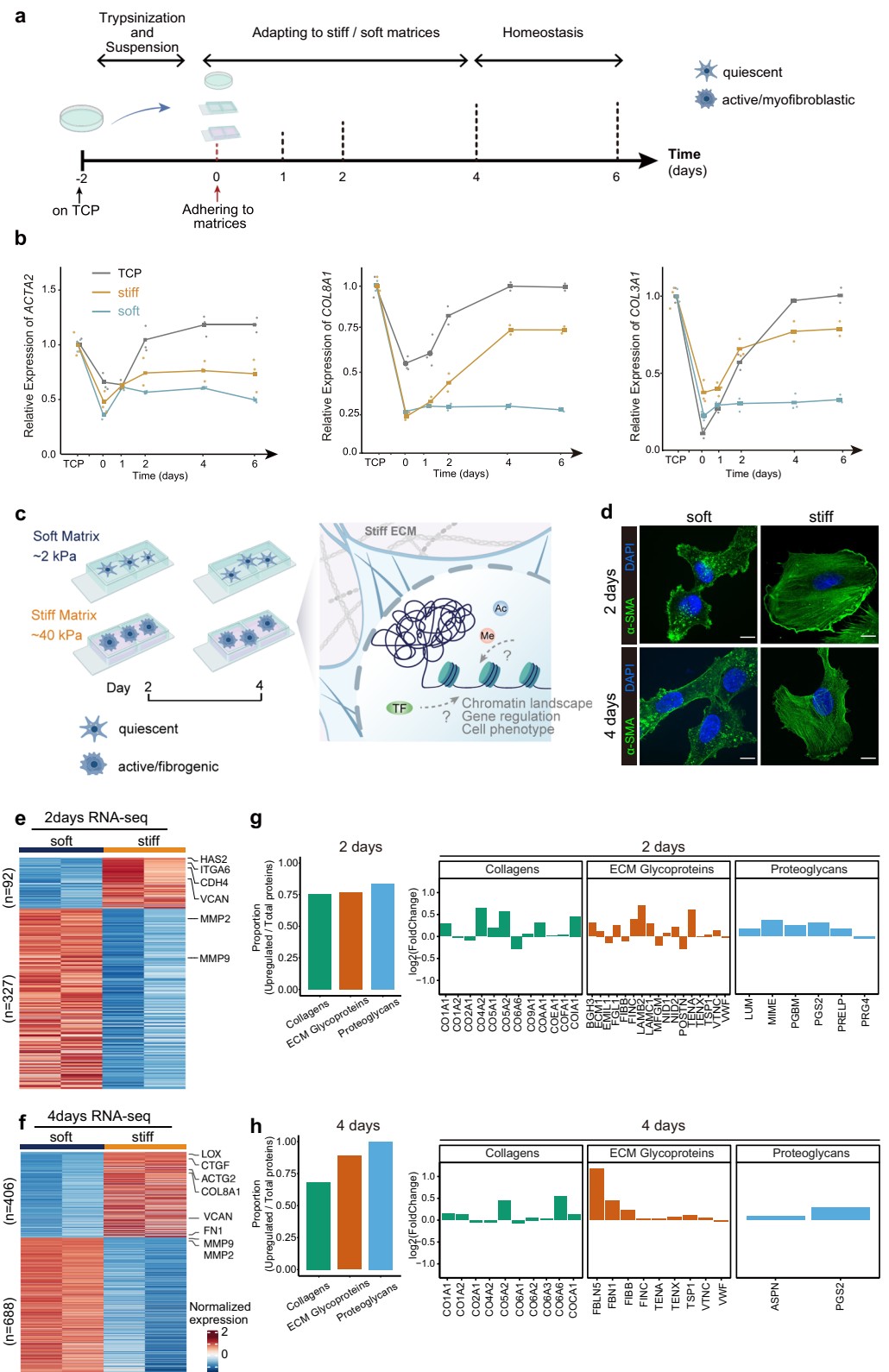

chromatin sites, which may be responsible for the upregulation of fibrosis-associated genes. To explore the manner by which DAPs induced by matrix stiffness affect the transcriptome, we characterized DAPs according to their relative locations to genes. DAPs were mostly located in intergenic and intronic regions, where enhancers were abundant and may play a key role in gene regulation (Fig. 2b)[31].

## Stiffness-induced dynamic chromatin accessibility precedes changes in gene expression

To better understand the relationship between chromatin accessibility and the regulation of fibrosis-associated genes in response to matrix stiffness, we compared the differentially accessible regions with gene expression patterns. We first identified the open chromatin regions unique to each sample and

**Fig. 1 | Different matrix stiffness facilitates HSC fibrogenesis as revealed by transcriptome profiling. a** Schematic of cell phenotypic transition process when being isolated and reseeded onto matrices. Created in BioRender. Zhao, W. (2025) https://BioRender.com/722zjib. **b** The relative expression levels of fibrotic genes in LX-2 cell lines over a period of days. The cells were initially plated on TCP and then isolated and reseeded onto soft and stiff matrices and TCP. Each set of data contains at least two independent replicates. The bolded horizontal lines in each condition indicated the mean values. **c** An overview of the study design depicting the workflow for cell culture, multi-omics analysis, imaging, and biochemical assays. This figure was created with BioRender.com. **d** Representative immunofluorescence images of LX-2 cells cultured on different matrices for two or four days. Scale bar, 10 μm. **e** Heatmaps of RNA-seq datasets for LX-2 cells cultured on different matrices for two days. Genes were grouped according to similar expression patterns, showing 92 significantly upregulated and 327 significantly downregulated genes. Marker

genes of fibrogenesis are listed on the right. Two independent experiments were performed for each category. **f** Heatmap of RNA-seq datasets for LX-2 cells cultured on different matrices for four days. Genes were grouped according to similar expression patterns, showing 406 significantly upregulated and 688 significantly downregulated genes. Marker genes of fibrogenesis are listed on the right. Two independent experiments were performed for each category. **g** The proportion of core ECM proteins that are upregulated in cells cultured on a stiff matrix for 2 days, relative to the total number of proteins detected. The expression changes of the three categories of core matrix proteins were detected in cells cultured for 2 days on stiff vs. soft matrices. **h** The proportion of core ECM proteins that are upregulated in cells cultured on a stiff matrix for 4 days, relative to the total number of proteins detected. The expression changes of the three categories of core matrix proteins were detected in cells cultured for 4 days on stiff vs. soft matrices.

then defined regions either shared between cells cultured for two and four days or between soft and stiff matrices, classifying them into eight clusters (Fig. 2c). As for the nomenclature for the clusters, the first four clusters represent regions that are specifically increased ATAC-seq sites under different culture conditions. For example, "D2 soft" represents the ATAC-seq sites that are specifically increased in cells cultured on the soft matrix for two days. While the last four clusters represent sites that are commonly increased under two of the culture conditions. For example, "D2 stiff + D4 stiff" represents the ATAC-seq sites that are commonly decreased both in cells cultured on the stiff matrix for two and four days (Fig. 2c). Subsequently, we evaluated the RNA expression patterns of genes adjacent to the chromatin regions in each cluster. As expected, the overall gene expression levels were positively correlated with promoter accessibility, for example, *IGFBP3* (Insulin-like Growth Factor Binding Protein 3) (Fig. 2c and Supplementary Fig. 2c), which is consistent with the notion that chromatin accessibility governs the expression of nearby genes[32]. Notably, a larger number of chromatin regions became more accessible in cells cultured for two days on stiff matrix relative to soft matrix; however, the genes adjacent to these regions were not upregulated until day 4 (Fig. 2d; the "D2 stiff + D4 stiff" cluster). Such regions contain several fibrosis marker genes such as *COL8A1* (Collagen Type VIII Alpha 1 Chain) (Fig. 2d). We defined these regions as mechanically primed chromatin regions, where dynamic chromatin accessibility precedes and foreshadows changes in gene expression in response to mechanical stimulus (Fig. 2e). Previous studies have reported that chromatin priming plays an important role in facilitating gene activation during cell differentiation and lineage specification, since it allows for a more rapid response to external signals[33,34]. Based on this evidence, the primed chromatin may regulate fibrosis-associated gene expression in response to mechanical signaling.

We next examined the enrichment of GO terms for genes adjacent to each cluster of primed chromatin regions. Although individual clusters were enriched in GO terms for various functions, most clusters were enriched in GO functions related to HSC activation in liver disease (Fig. 2c)[35,36]. Notably, the genes associated with primed chromatin regions were enriched in functions related to actin cytoskeleton organization, response to mechanical stimulus, and mesenchymal-to-epithelial transition, which have been reported in activated HSCs[37,38]. In addition, the genes that became more accessible in cells cultured on a stiff matrix for two days (the "D2 stiff" cluster) tended to participate in housekeeping kinase signaling pathways essential to the early stages of cell proliferation (Fig. 2c). However, the genes that became more accessible in cells cultured on a stiff matrix for four days (the "D4 stiff" cluster) were related to liver development and membrane protein proteolysis (Fig. 2c)[39], which have been reported to be activated during fibrotic progression[40,41], indicating that the activation of mechanical signaling transduction pathways affects cell behavior. The accessible peaks shared by cells cultured on the soft and stiff matrix for two days were enriched in cell migration and cytoskeletal organization pathways that contribute to cell spread and colonization on hydrogels (Fig. 2c). By contrast, the peaks shared by cells on day 4 were enriched in adherens junction organization and epithelial transition pathways related to increased cell

density (Fig. 2c). Collectively, dynamic chromatin accessibility induced by the stiff matrix is associated with diverse functional pathways converging to mechanosensing genes in response to mechanical stimulus.

## Primed chromatin accessibility remodeling is positively associated with activation of the H3K27ac modification

To explore the mechanism underlying the establishment of primed chromatin, we characterized the genomic features of the primed chromatin regions and subsequently employed chromHMM to train 10 chromatin states of HSCs using five epigenetic markers[42]. The chromatin states were trained using ENCODE ChIP-seq data generated from primary HSCs. Previous studies have reported that changes in histone modifications may precede changes in gene expression, resulting in the establishment of primed chromatin[43]. We found that primed chromatin regions were enriched in bivalent promoters (Fig. 3a) containing both H3K27me3 and H3K4me3, representing repressive and activating histone modifications that maintain these chromatin regions poised for activation[43]. Moreover, primed chromatin regions were also enriched in active enhancers (Fig. 3a) containing H3K27ac, which are responsible for enhancer activation[44]. These results are in accordance with those previously reported that H3K27ac modifications may mark genomic regions primed for activation[45].

To further explore the association between histone modifications and the establishment of primed chromatin in response to matrix stiffness, we performed Cut&Tag experiments targeting H3K4me3, H3K27me3, and H3K27ac in LX-2 cells cultured for two days on soft and stiff matrices, and compared the changes in histone modifications among the 8 chromatin accessibility clusters[46]. The results showed a globally similar distribution of histone modifications in the whole genome-wide (Fig. 3b and Supplementary Fig. 3a). The H3K4me3 and H3K27me3 levels remained stable in all the clusters in response to the stiff matrix (Fig. 3c). By contrast, the H3K27ac modification level was specifically increased in the "primed chromatin" and "D2 stiff" clusters but not in the other clusters, indicating specifically increased chromatin accessibility in these regions in response to the stiff matrix (Fig. 3c). As an example, the fibrosis marker gene *TIMP4* displayed increased chromatin accessibility and H3K27ac modification levels; however, there was no change in the histone methylation levels within its promoter region (Fig. 3d). This finding is consistent with previous reports that H3K27ac contributes to chromatin accessibility and distinguishes active from inactive regulatory elements[44]. Taken together, these results suggest that H3K27ac is involved in the process by which stiffness-induced primed chromatin is prepared for transcriptional activation.

## AP-1 factor motifs are enriched in primed chromatin regions in response to matrix stiffness

The universality of priming mechanisms that include both selective histone modifications and transcription factor (TF) binding has been confirmed by genome-wide studies[47,48]. Moreover, one study focusing on chromatin reprogramming and genome activation revealed that pioneer factors can bind nucleosomal DNA to regulate histone modifications and chromatin accessibility during zygotic genome activation (ZGA), suggesting that

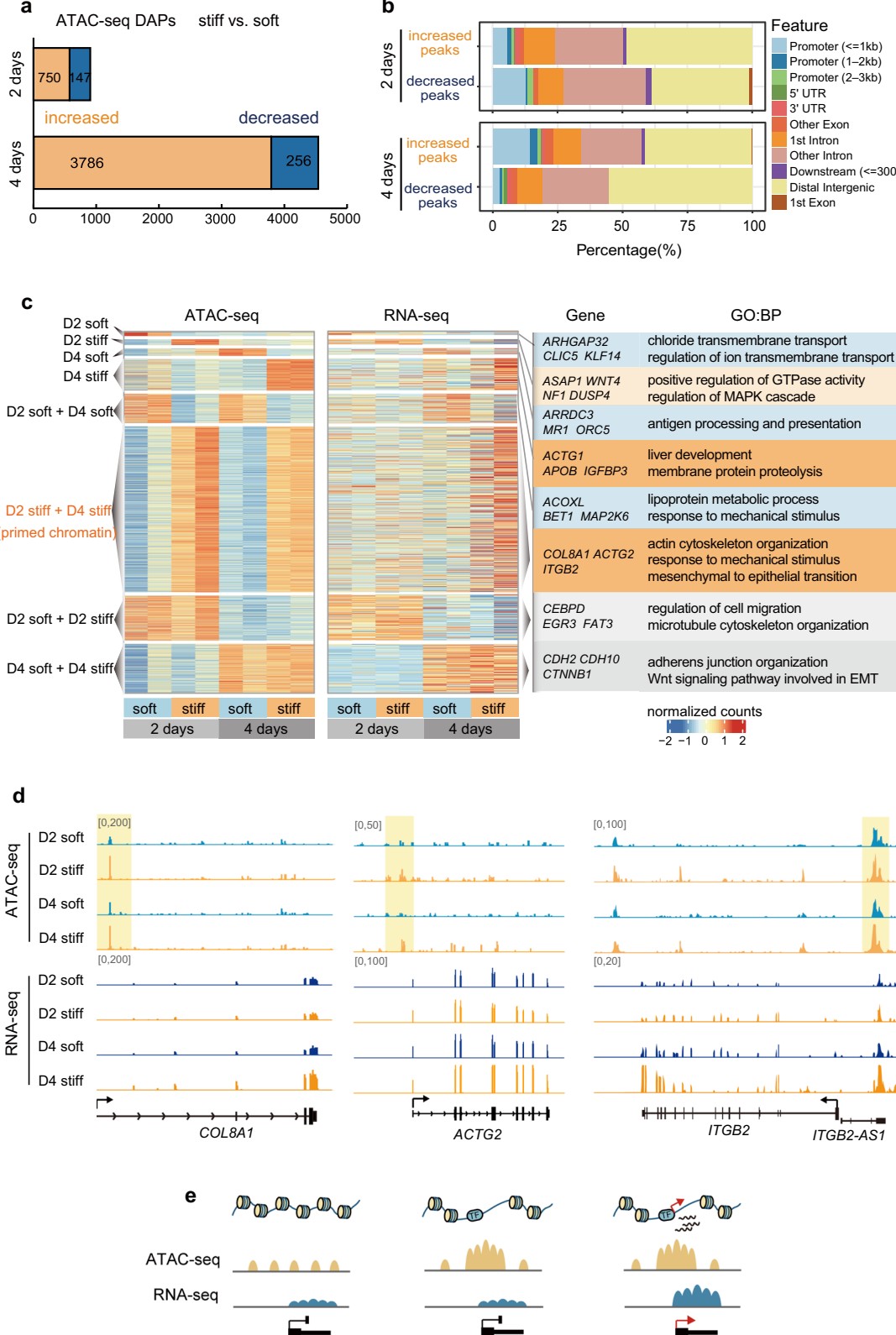

**Fig. 2 | Dynamics of chromatin accessibility during fibrogenesis. a** The stack bar plot showing the differentially accessible peaks (DAPs) for cells cultured on different matrices for two or four days. **b** Genomic feature annotation of differentially accessible ATAC-seq peaks. **c** Analysis of unique or shared clusters of open chromatin regions. Heatmap shows ATAC-seq signals for unique and shared peak groups and adjacent gene RNA-seq patterns for echo groups (left panel). The marker genes and top GO terms for each cluster are listed on the right (right panel). **d** Normalized ATAC-seq signal profiles of primed chromatin loci in *COL8A1, ACTG2, ITGB2* from cells on the soft or stiff matrix, shown together with a normalized RNA-seq profile by the sequencing depth. The transcription start site of each gene is marked by yellow box. **e** Cartoon model showing that changes in primed chromatin accessibility foreshadow changes in gene expression.

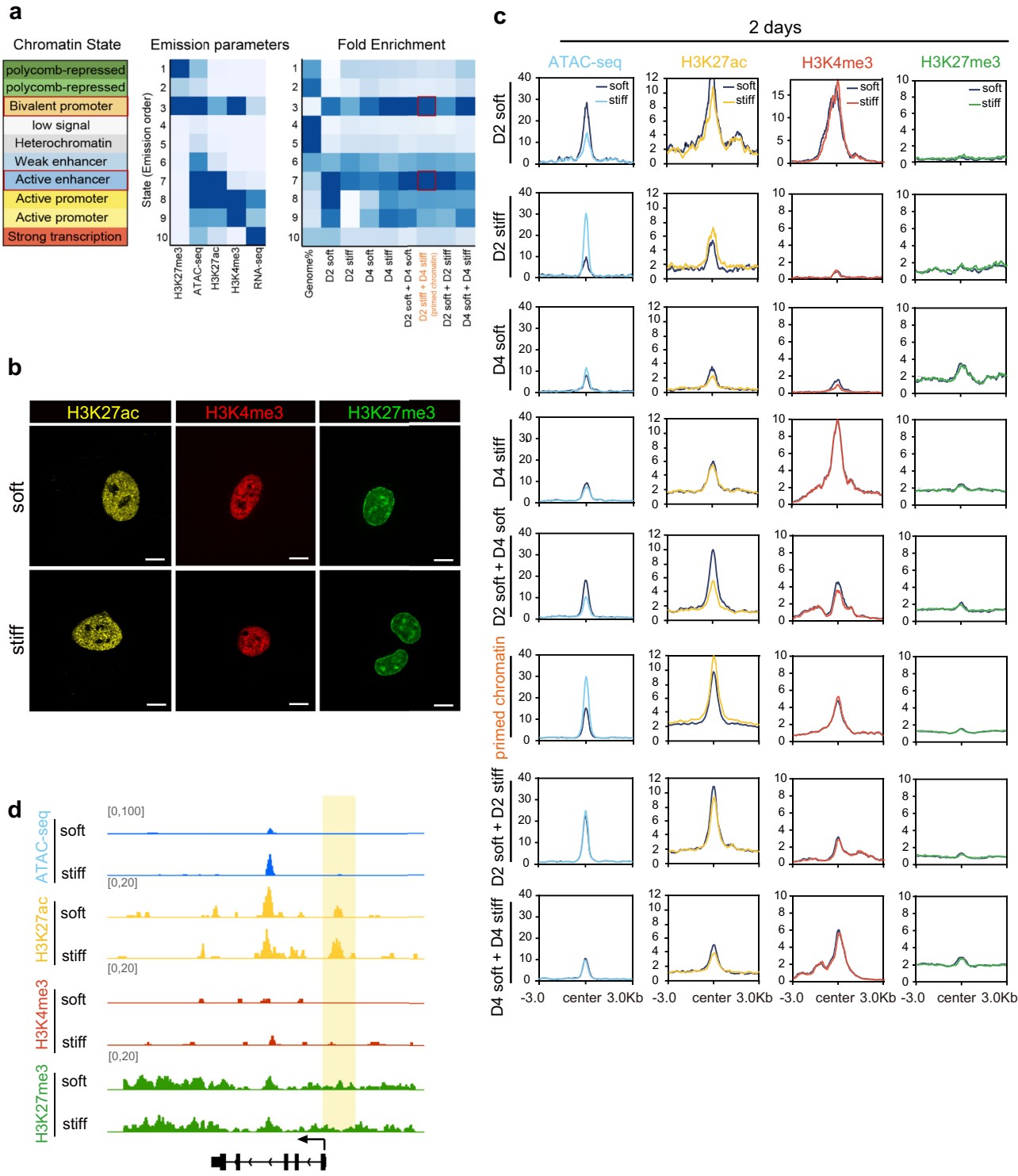

**Fig. 3 | Characterization of dynamic histone modifications around primed chromatin. a** Ten chromatin markers were used to train the chromatin states in primary hepatic stellate cells by chromHMM. The different shades of the same color were used to distinguish the chromatin state. The middle section displaying a heatmap of the emission parameters, in which each row corresponds to a different state, and each column corresponds to a different mark defined on the basis of the observed data. The right heatmap displays the fold enrichment for the eight chromatin accessibility clusters in epigenomic-marked chromatin states. A darker blue color corresponds to a greater fold enrichment, and there is one color scale for the entire heatmap. **b** Profiles of H3K4me3, H3K27me3, and H3K27ac signals surrounding different chromatin clusters as defined in Fig. 2a. Rows correspond to ±3 kb regions across the midpoint of each primed chromatin site. **c** Normalized H3K4me3, H3K27me3, and H3K27ac signal profiles at a locus in *TIMP4* are shown together with a normalized ATAC-seq profile. The vertical yellow boxes highlight the transcription start sites. **d** Representative immunofluorescence images of cells cultured on different matrices for two days. Anti-H3K27ac (yellow), anti-H3K4me3 (red), and anti-H3K27ac (green). Scale bar, 10 μm.

H3K27ac is not absolutely required for chromatin opening[49]. Furthermore, TFs are regarded as master regulators of the epigenome and global transcriptional activity[50,51]. To explore potential TFs responsible for the primed chromatin regions induced by a stiff matrix, we performed HOMER enrichment analysis to identify enriched TF binding motifs[52]. The activator protein-1 (AP-1) and TEAD families were the most enriched motifs in primed chromatin regions (Fig. 4a). TFs in the AP-1 family bind DNA and contribute to genome-wide remodeling of the epigenetic landscape, and thus may function as pioneer factors[53–55]. In support of this role, the AP-1 motif had stronger footprint signals in primed chromatin regions of cells cultured on the stiff matrix (Fig. 4b). These results suggest that AP-1 family TFs may be responsible for primed chromatin regions and downstream changes in gene expression.

Next, we asked the question whether the increased mRNA expression or enhanced binding of activated AP-1 family TFs contributes to primed chromatin remodeling in response to matrix stiffness. As the previous studies reported, AP-1 is a menagerie of dimeric basic region-leucine zipper (bZIP) proteins that belong to the Jun (c-Jun, JunB, JunD), Fos (c-Fos, FosB, Fra-1 and Fra2), Maf (c-Maf, MafB, MafA, MafG/F/K and Nrl) and ATF (ATF2, LRF1/ATF3, B-ATF, JDP1, JDP2) sub-families, each sub-family of TFs share some similarities in structure and function[56,57]. RNA-seq shows that the expression of most AP-1 TFs remained stable in cells cultured on the soft and stiff matrices (Fig. 4c). The scRNA-seq using primary HSCs from patients with liver cirrhosis and healthy donors[36] showed that the transcription level of *JUN* was moderately decreased in cirrhotic livers (Supplementary Fig. 4a–d). Taken together, these results suggested that *JUN* expression remained stable, and the mRNA expression level was not the main reason for its enhanced enrichment on primed chromatin in response to the stiff matrix.

Although TF expression provides the first line of evidence for the locations in which they may function, their activity often depends on post-transcriptional events, and the mRNA levels do not necessarily indicate regulatory activity[58]. As TFs can move around and bind primed chromatin without necessarily increased expression, we next focused on exploring whether the binding of activated AP-1 TFs was enhanced, resulting in primed chromatin state. Since some of the AP-1 TFs (JUN, FOS) can be phosphorylated at distinct residues by different kinases and thus possess enhanced transcriptional activity[59], we compared the change of protein levels of JUN, p-JUN, FOS, p-FOS, and ATF3 by western blot using specific antibodies (Table 1). We found that p-JUN and ATF3 expressed at higher protein levels in cells cultured on a stiff matrix for two days, while other factors in the AP-1 family remained stable at the protein level (Fig. 4d). Next, we used the CUT&Tag technique targeting p-JUN and ATF3 in the same experimental batch to capture its binding affinity in the different clusters of chromatin regions. An increased binding level of p-JUN was found specifically in primed chromatin regions (Fig. 4e), and the fraction of p-JUN on accessible peaks also showed a preference for primed chromatin regions (Fig. 4f). However, the enrichment of ATF3 in primed chromatin regions remains unchanged in response to matrix stiffness (Supplementary Fig. 5a). Additionally, the fraction of ATF3 on accessible peaks does not show a preference for primed chromatin regions like p-JUN (Supplementary Fig. 5b). JUN is the most efficacious transcriptional activator in the AP-1 family[56] and is considered an important regulator of hepatic stress responses[60]. Taken together, these results indicate that AP-1 factors, in particular p-JUN, are enriched in primed chromatin regions and may mediate their establishment in response to mechanical stimulus.

## Phosphorylation of JUN promotes chromatin accessibility in response to matrix stiffness

To further explore the mechanism underlying the regulation of chromatin accessibility by p-JUN, we designed plasmids containing mutated phosphorylation sites at Ser[61] and Ser[62] within JUN to achieve enhancement (the JUN-EE mutant) of its transcriptional activation. Moreover, we also designed the JUN-EE-Δbasic and JUN-EE-Δleucine mutants to interfere with the DNA-binding activity of JUN and its dimerization with other TFs,

respectively (Fig. 5a)[63]. To quantify the phosphorylation level of JUN mutants (JUN-EE, JUN-AA), we utilized the normalized fluorescent intensity to compare the relative expression levels of p-JUN in cells. The results showed that the phosphorylation level of cells with JUN-EE mutant was approximately 3-fold higher than that in control cells, while the phosphorylation level of JUN-AA mutant was about 3-fold lower than control cells (Fig. 5b, c), which is consistent with the expected functions of JUN mutants[63].

After that, ATAC-seq was performed to identify changes in chromatin accessibility. Enhanced phosphorylation of JUN (the JUN-EE mutant) resulted in increased chromatin accessibility in the primed chromatin regions of cells cultured on a soft matrix (Fig. 5d). By contrast, the JUN-EE-Δbasic and JUN-EE-Δleucine mutants resulted in reduced chromatin accessibility as compared with JUN-EE and caused the chromatin accessibility similar to that of control cells (Fig. 5d), which was in line with the mechanism that the dimerization and DNA binding activity were required for p-JUN to perform its transcription function[56]. Accordingly, we also compared the ATAC-seq signal in the other chromatin regions excluding primed chromatin, and found that the chromatin accessibility increased moderately, suggesting a specific function of p-JUN in the establishment of primed chromatin (Supplementary Fig. 4e). We then investigated whether p-JUN is indispensable for chromatin accessibility in response to matrix stiffness. The deficient phosphorylation of JUN (the JUN-AA mutant) resulted in decreased chromatin accessibility in the primed chromatin regions of cells cultured on a stiff matrix (Fig. 5e). Accordingly, when comparing the ATAC-seq signal in the other chromatin regions excluding primed chromatin, we also found decreased chromatin accessibility (Supplementary Fig. 4f). These results indicate that p-JUN is required for the establishment of accessible chromatin globally.

Furthermore, we examined the transcriptional changes of fibrotic genes upon the gain and loss of function of p-JUN by day 4, the results showed that the expression of the fibrosis marker α-SMA was increased in cells with enhanced p-JUN, while its expression was decreased in cells with deficient p-JUN (Fig. 5f). Immunofluorescence imaging shows that the cells containing the JUN-EE mutant had well-organized actin stress fibers compared with the control cells expressing wild-type JUN, conversely, the JUN-AA resulted in deficient stress fiber organization (Fig. 5g). These results indicated that the phosphorylation of JUN could promote the accessibility of primed chromatin and the phenotypic shift towards fibrosis.

Next, we explored whether or not p-JUN is elevated in HSCs during liver fibrosis. We conducted immunohistochemistry (IHC) experiments using CCl₄-treated chronic fibrosis mouse. Masson's trichrome staining results showed that the liver tissue had significantly fibrosed after 6 weeks of CCl₄ injection (Fig. 5h, i). On consecutive sections of fibrotic liver and control liver, activated HSCs were marked with α-SMA antibody, and the p-JUN was also labeled. The results showed an increase in p-JUN protein expression in HSCs after fibrosis (Fig. 5j, k). This indicates that the phosphorylation and activation of the TF JUN may play an important role in the fibrotic process in vivo.

Given that the phosphorylation of JUN regulates chromatin dynamics under mechanical stimulation, we further sought to elucidate the upstream signaling pathways responsible for the changes in ECM stiffness and regulation of TF phosphorylation levels. It has been suggested that ECM stiffness may regulate the activities of cytoplasmic kinases and subsequently influence cell behavior[64]. The conserved extracellular signal-regulated kinase (ERK) signaling pathway is activated by a variety of extracellular stimulus[61,65]. To address the participation of upstream signaling pathways after the application of matrix stiffness, we performed immunofluorescence assays targeting p-ERK. We found that p-ERK had a higher nuclear-to-cytoplasmic fluorescence ratio in response to a stiff matrix (Supplementary Fig. 6a, b). Next, we used specific inhibitors of ERK, JNK kinases, and p38, which have been reported to be important in the response to stimulus and tumorigenesis (Supplementary Fig. 6c, d). Lower p-JUN levels were observed following inhibition of p-ERK, while the inhibition of other kinases or pathways had no effect, suggesting that p-ERK is critical for the

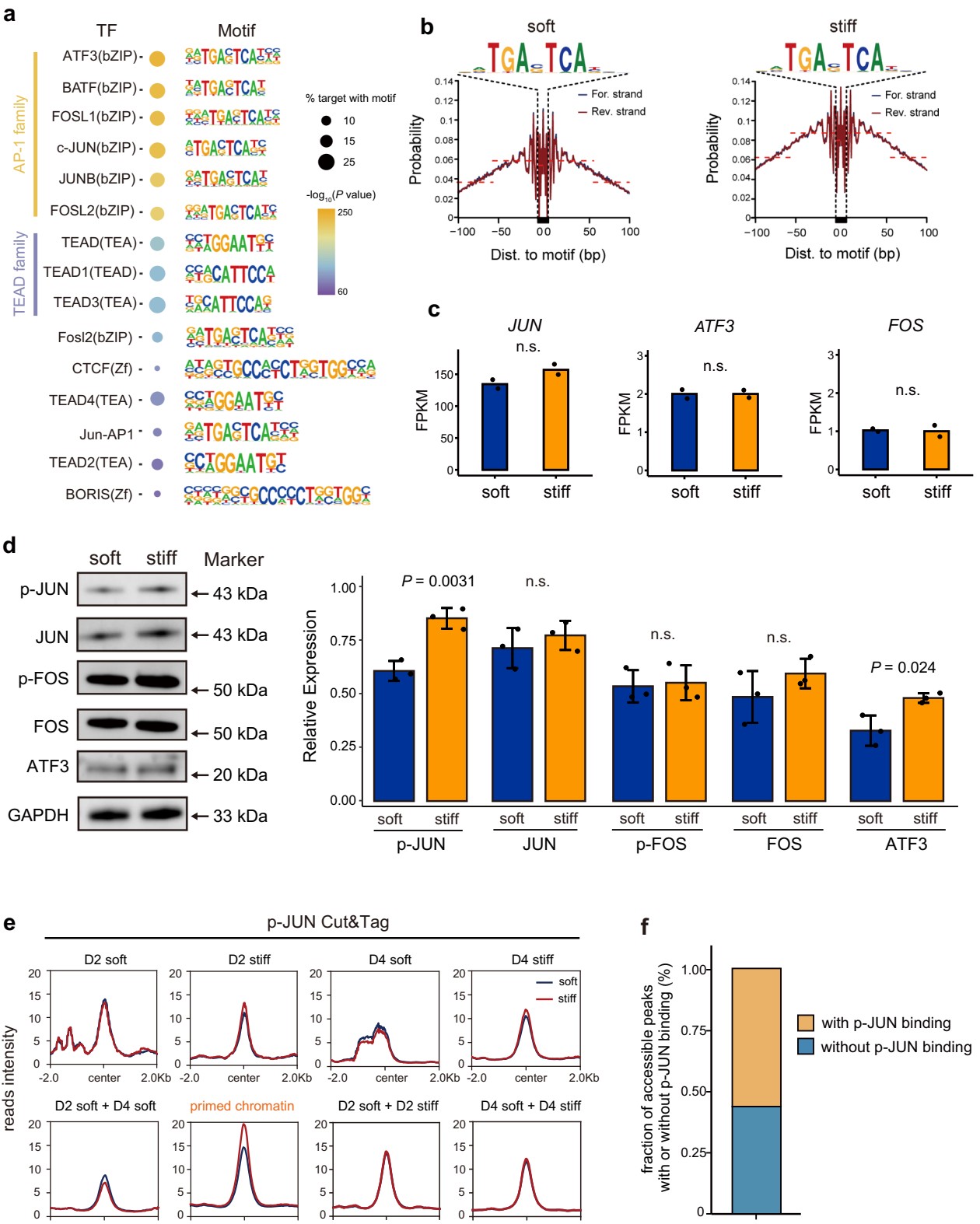

**Fig. 4 | AP-1 factor motifs are enriched in primed chromatin regions in response to matrix stiffness. a** Transcription factor enrichment in primed chromatin regions. **b** ATAC-seq footprint showing motifs enriched in primed chromatin regions from cells in the soft and stiff matrices. **c** mRNA expression levels of *JUN*, *ATF3*, *FOS*, *JUNB*, *FOSL1*, and *FOSL2* in LX-2 cells cultured on soft and stiff matrix for two days (n.s. non-significant, two-tailed Wilcoxon test; two independent experiments). **d** Western blots and statistical results showing the protein expression levels of p-Jun in LX-2 cells cultured for two days on the soft and stiff matrices. GAPDH served as the loading control. Western-blot are representative of three independent experiments. The statistical results are shown as mean ± standard deviation (SD), one-way ANOVA. **e** CUT&Tag signal showing the binding of p-JUN in the different chromatin clusters defined in Fig. 2c. **f** Stacked bar plots showing the binding faction of p-JUN in defined chromatin region clusters. Primed chromatin is highlighted in yellow.

## Table 1 | Resources

| Reagent or resource | Source | Identifier |
|---|---|---|
| **Antibodies** | | |
| Rabbit anti-SMA | Abways | CY5295 |
| Rabbit anti-P-FOS | Cell signaling technology | 5348T |
| Rabbit anti-P-JUN | Cell signaling technology | 3270T |
| Rabbit anti-ATF3 | Abcam | ab254268 |
| Mouse anti-c-JUN | Santa Cruz Biotechnology | sc-74543 |
| Mouse anti-c-FOS | Santa Cruz Biotechnology | sc-166940 |
| Mouse anti-GAPDH | Abcam | Ab8245 |
| Rabbit anti-H3K27ac | Abcam | ab4729 |
| Rabbit anti-H3K4me3 | Abcam | ab8580 |
| Mouse anti-H3K27me3 | Abcam | ab6002 |
| **Biological samples** | | |
| Goat anti-rabbit IgG HRP | EASYBIO | BE0101 |
| Goat anti-Mouse IgG HRP | EASYBIO | BE0102 |
| Alexa Fluor 594 donkey anti-mouse | Thermo Fisher Scientific | A21208 |
| Alexa Fluor 594 donkey anti-rabbit | Thermo Fisher Scientific | A21207 |
| Alexa Fluor 488 donkey anti-rabbit | Thermo Fisher Scientific | A21206 |
| Alexa Fluor 647 goat anti-mouse | Thermo Fisher Scientific | A21235 |
| Alexa Fluor 488 donkey anti-mouse | Thermo Fisher Scientific | A21202 |
| **Chemicals, peptides, and recombinant proteins** | | |
| TruePrep DNA library prep kit V2 | Vazyme | TD502 |
| HiScript® II 1st strand cDNA synthesis Kit | Vazyme | R211-01 |
| NovoNGS® CUT&Tag 3.0 high-sensitivity kit | NovoProtein | N259-YH01 |
| Nuclear and cytoplasmic protein extraction kit | YEASEN | 20126ES50 |
| MolPure® cell RNA kit | YEASEN | 19231ES50 |
| ERK1/2 inhibitor | MedChemExpress | PD98059 |
| JNK1/2/3 inhibitor | MedChemExpress | SP600125 |
| P38 inhibitor | MedChemExpress | SB203580 |
| **Deposited data** | | |
| RNA-seq, ATAC-seq, CUT&Tag | This paper | |
| ChIP-seq | ENCODE | ENCFF781SNK, ENCFF050OMW, ENCFF760ABV, ENCFF064AZN, ENCFF055ACA, ENCFF539DCQ, ENCFF414RVB |
| **Experimental models: Cell lines** | | |
| Human LX-2 cells | Procell | CL-0560 |
| **Oligonucleotides** | | |
| The primers used for qPCR can be found in the supplementary information. | This paper | |
| **Software and algorithms** | | |
| Bowtie2 | Langmead and Salzberg[109] | https://bowtie-bio.sourceforge.net/bowtie2/index.shtml |
| MACS2 | Zhang et al.[110] | https://github.com/taoliu/MACS |
| Homer | Heinz et al.[52] | http://homer.ucsd.edu/homer/motif/ |
| STAR | Dobin et al.[113] | https://github.com/alexdobin/STAR |
| DESeq2 | Love et al.[104] | http://www.bioconductor.org/packages/release/bioc/html/DESeq2.html |
| DiffBind | Stark and Brown (2011) | https://bioconductor.org/packages/release/bioc/html/DiffBind.html |
| Samtools | Danecek et al.[114] | http://samtools.sourceforge.net |
| Trimgalore | N/A | http://www.bioinformatics.babraham.ac.uk/projects/trim_galore |
| R | N/A | https://www.rproject.org/ |
| ImageJ | N/A | https://imagej.net/ij/index.html |
| IGV | Robinson et al. (2011) | https://www.igv.org/ |
| chromHMM | Ernst and Kellis,[42] | https://compbio.mit.edu/ChromHMM/ |
| PrimerBank | Wang and Seed[115] | https://pga.mgh.harvard.edu/primerbank/ |

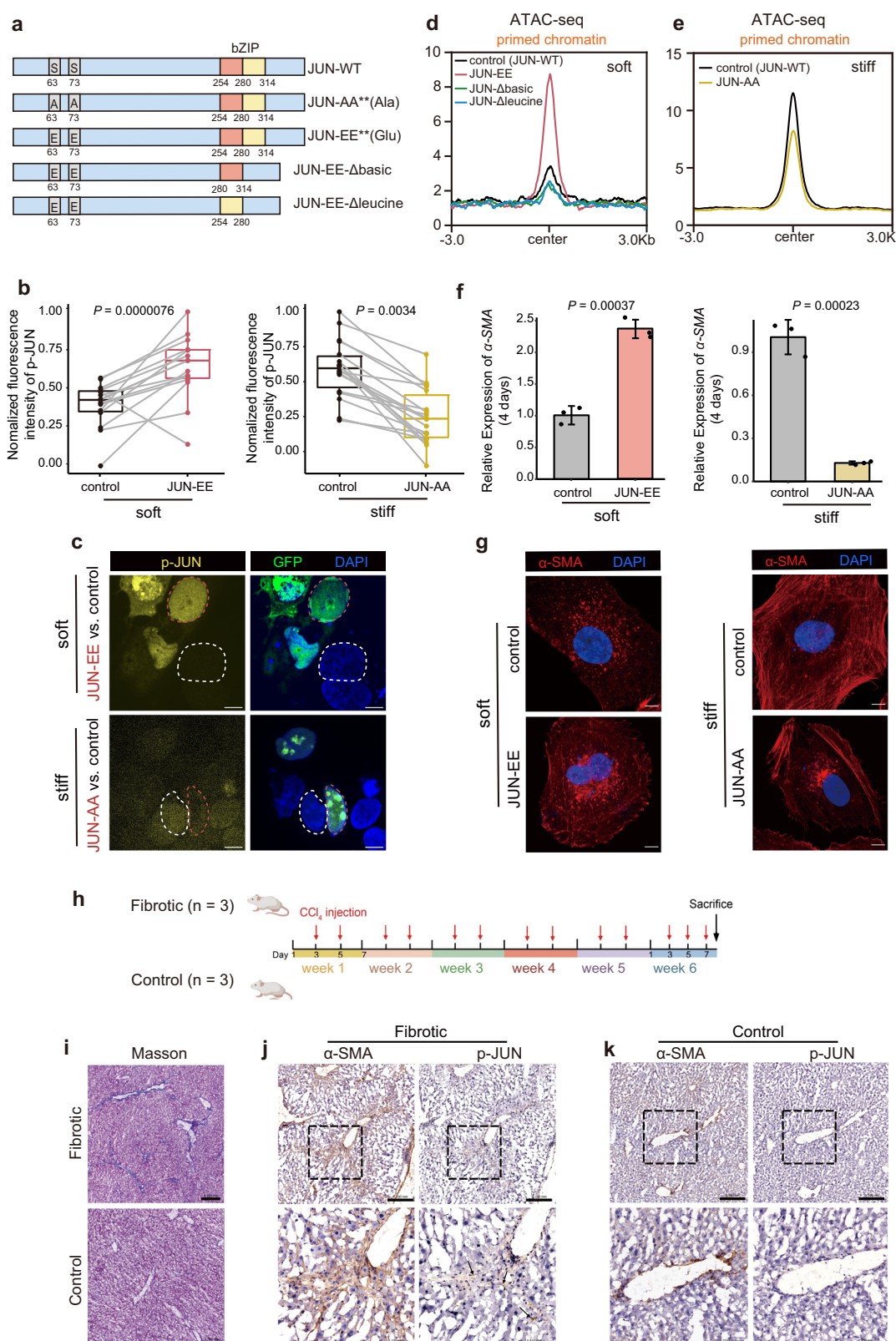

activation of JUN. These results indicate that the ERK pathway acts as a molecular link between matrix stiffness and chromatin accessibility, which in turn enables cells to dynamically regulate gene expression and adapt to altered mechanical environments. Moreover, we continuously observed the nuclear-cytoplasmic fluorescence ratio of p-ERK on the stiff matrix to explore the timing and duration of p-ERK activity. The results showed that

the activation of ERK responded to stiffness in a relatively short period of time (within 2 h) and maintained the activated state for a long time (Supplementary Fig. 6e, f). Previous publications reported that the translocation of ERK can be either transient, with a peak at around 10–15 min after stimulation, or prolonged, where most ERK molecules enter the nucleus within 10 min and can be retained there by various nuclear anchoring

**Fig. 5 | Activation of p-Jun through the ERK signaling pathway contributes to the maintenance of primed chromatin accessibility. a** Schematic structure of the JUN protein sequence showing the DNA-binding and dimerization domains of JUN (bZIP) and the JNK phosphorylation sites (Ser[63] and Ser[73]). For JUN-AA, point mutations were introduced to convert Ser[63] and Ser[73] to Ala. For JUN-EE, point mutations were introduced to convert Ser[63] and Ser[73] to Glu. For JUN-EE-Δbasic, the DNA-binding domain was truncated based on JUN-EE. For JUN-EE-Δleucine, the heterodimerization domain with other AP-1 family members was truncated based on JUN-EE. **b** Normalized mean fluorescence intensity of immunostaining signals of p-JUN in cells cultured on the soft matrix with or without JUN-EE mutant, and the signals of p-JUN in cells cultured on the stiff matrix with or without JUN-AA mutant. signals of p-JUN in cells cultured on the soft matrix with or without JUN-EE mutant. About 15 pairs of cells within the same imaging view were analyzed for each independent experiment. Paired Wilcoxon test. **c** Immunostaining of p-JUN showing the fluorescence intensity of p-JUN induced by the mutant. The merged image of DAPI and GFP indicating the cell expressed mutant JUN-AA or JUN-AA (outlined by red circle) and the control cell (outlined by white circle). Scale bars, 5 μm. **d** Normalized ATAC-seq signal profiles in the primed chromatin regions for cells cultured on the soft matrix following expression of JUN-EE, JUN-EE-Δbasic, and JUN-EE-Δleucine are shown in (**a**). **e** Normalized ATAC-seq signal profiles in the primed chromatin regions for cells cultured on the stiff matrix following expression of JUN-AA are shown in Fig. 5a. **f** Relative mRNA expression of α-SMA on the soft or stiff matrices for 4 days with or without mutants JUN. The expression level was normalized with GAPDH, three independent experiments, the separate one-way ANOVA analysis was done in each figure. The results are shown as mean ± SD. **g** Immunofluorescence images of soft matrix-induced cytoskeletal structures reconstructed following expression of JUN-EE. Scale bar, 10 μm. And Immunofluorescence images of stiff matrix-induced cytoskeletal structures reconstructed following expression of JUN-AA Scale bar, 10 μm. The image was collected from cells on different matrices for 4 days. **h** Mouse model of liver fibrosis induced by CCl₄ administration and the CCl₄ treatment schedule. $n = 3$ biologically independent samples per group. Created in BioRender. Zhao, W. (2025) https://BioRender.com/j7b1xe3. **i** Visualization of the degree of liver fibrosis by Masson's trichrome staining, where collagen fibers are stained blue. Scale bar, 200 μm. IHC characterizes the expression levels of p-JUN in fibrotic mouse liver cells. Consecutive sections from fibrotic liver (**j**) and control liver (**k**) are stained with α-SMA antibody to mark activated hepatic stellate cells, and with the p-JUN. Scale bar, 200 μm. Enlarged views of the areas enclosed by the dashed boxes are shown below, scale bar, 100 μm. The black arrows in the images emphasize the regions where hepatic stellate cells are located.

moieties for up to 3 h or even longer (>10 h)[66,67]. Based on these, we proposed that the matrix stiffness can be regarded as continuous and long-term extracellular stimulus to the cell, thereby leading to a sustained ERK activity responding dynamics.

## Discussion

Here, we demonstrate that ECM stiffness induces cell phenotypic transition through the remodeling of chromatin accessibility. Moreover, we show that the establishment of primed chromatin in response to the matrix mechanical stimuli precedes the onset of gene transcription. Overall, based on our work and prior knowledge, we propose a model underlying the regulation of gene expression in response to mechanical stimulus. At the first stage, the cells are subjected to the increased matrix stiffness on a shorter time scale, the cytoplasmic kinases ERK would be activated and translocate to the nucleus, and then phosphorylate the pioneer factor (JUN), leading to its transcriptional activation state (p-JUN) and promoting its binding to chromatin. Subsequently, the pioneer factor can initiate chromatin opening to establish a primed chromatin state. During this process, we find that the H3K27ac modification is increased, which is consistent with the previous report that the pioneer factor can facilitate the recruitment of the general transcription co-activator p300, CREB-binding protein (CBP), and acetylated H3K27[51]. However, the nearby genes are not yet transcriptionally activated. At the second stage, when the cells are subjected to the stiff matrix stimuli on a longer time scale, the genes near the pioneer factor start to be activated and expressed, and the cells exhibit a pronounced fibrotic phenotype (Fig. 6). In summary, our findings emphasize a major role for chromatin priming in the response to ECM mechanical stimuli, which sheds light on the gene regulation mechanism in cells cultured on a stiff matrix. Given that both histone acetylation and pioneer factors are involved in dynamic chromatin accessibility, the detailed functions of AP-1 factors and histone acetyltransferases (HATs) in the regulation of the chromatin landscape remain unclear.

Previous publications have discovered the contribution of ECM stiffness to cell phenotypic transition through transcriptome alterations[2,68,69]. Moreover, the chromatin landscape can also be affected by matrix stiffness; for example, changes in matrix stiffness can drive changes to the nucleus and chromatin state through Sp1-HDAC3/8-mediated pathways, in which Sp1 can recruit HDACs (histone deacetylases) to further modulate chromatin accessibility[4]. The transcriptional regulation controlled by epigenetics, such as DNA methylation, histone modifications, has been widely explored in the context of HSCs activation and the mechanical activation of other mesenchymal cells. Epigenetic events that control HSC activation and function are highly dynamic processes[70], and several relevant epigenetic mechanisms have been reported[71]. Tian et al. reported that Myocardin-related TF A (MRTF-A, also known as MKL/myocardin-like protein 1) orchestrates profibrogenic transcription by recruiting a histone methyltransferase complex to the promoters of fibrogenic genes by regulating the H3K4 methylation[72]. In our work, we defined the primed chromatin regions, where the chromatin accessibility precedes and foreshadows changes in gene expression.

Evaluation of genome-wide chromatin accessibility using ATAC-seq of cells stimulated by mechanical cues (including stiffness, stretch, and topography) revealed a dramatic effect on the global chromatin structure[4,62,73–76]. However, the chromatin structure of cells does not always follow similar trends in chromatin accessibility in response to micro-environmental stiffness cues. For instance, one previous study found that mesenchymal stem cells (MSCs) maintain a more accessible chromatin status on soft substrates than those on stiff substrates[76], while other works reported that stiff matrix resulted in more accessible chromatin of fibroblast2 and mammary epithelial cells[4]. Our results showed that stiff matrix induced the establishment of primed chromatin in HSCs, in which the chromatin regions become more accessible prior to the upregulation of nearby genes. Since HSCs have also been described as a liver-resident MSCs population and can transdifferentiate to myofibroblasts[77,78], it is interesting to find that these cell types have distinct mechanisms regulating chromatin accessibility upon mechanical stimulus, indicating that there are still many issues worth studying between cell fate and the response to mechanical signals. Moreover, previous studies have reported that increased expression levels of the JUN gene in fibrotic diseases may be highly correlated with the disease[60,79]. The upregulation of TF or their phosphorylation and activation both have an impact on their transcriptional activation capacity[80]. Our research found and demonstrated that the TF JUN did not show significant changes at the gene and protein expression levels; instead, its phosphorylation and activation status responded to mechanical signals. Additionally, scRNA-seq data from patients with liver cirrhosis also revealed no changes in JUN gene expression. Our results indicated the impact of the phosphorylation and activation of TF on their transcriptional activity.

Epigenetic regulators and TF can also organize the genome into accessible or closed regions, orchestrating the appropriate transcriptional program in the cell[81]. Many previous studies have reported that YAP1 is a critical driver of HSC activation, as its nuclear translocation is followed by the induction of YAP1 target genes during activation[82–84], and pharmacological inhibition of YAP1 improves liver fibrosis in mice[85]. Jang et al., using the human gastric cancer cell line, revealed that the DNA methylation of the promoter region of the oncogenic YAP is reversibly regulated by the stiffness of the ECM[5]. Additionally, TEAD proteins have been shown to interact with YAP/TAZ, a key component of the Hippo signaling pathway, could sense the physical nature and drive tumorigenesis[86,87]. In our work, we found that

**Fig. 6 | A schematic illustrating the mechanism underlying cell phenotype shifts in response to matrix stiffness.** The proposed model shows that the stiff matrix induces cell phenotype transition, and the chromatin accessibility dynamics occur before transcription activation. Created in BioRender. Zhao, W. (2025) https://BioRender.com/y5ixhxi.

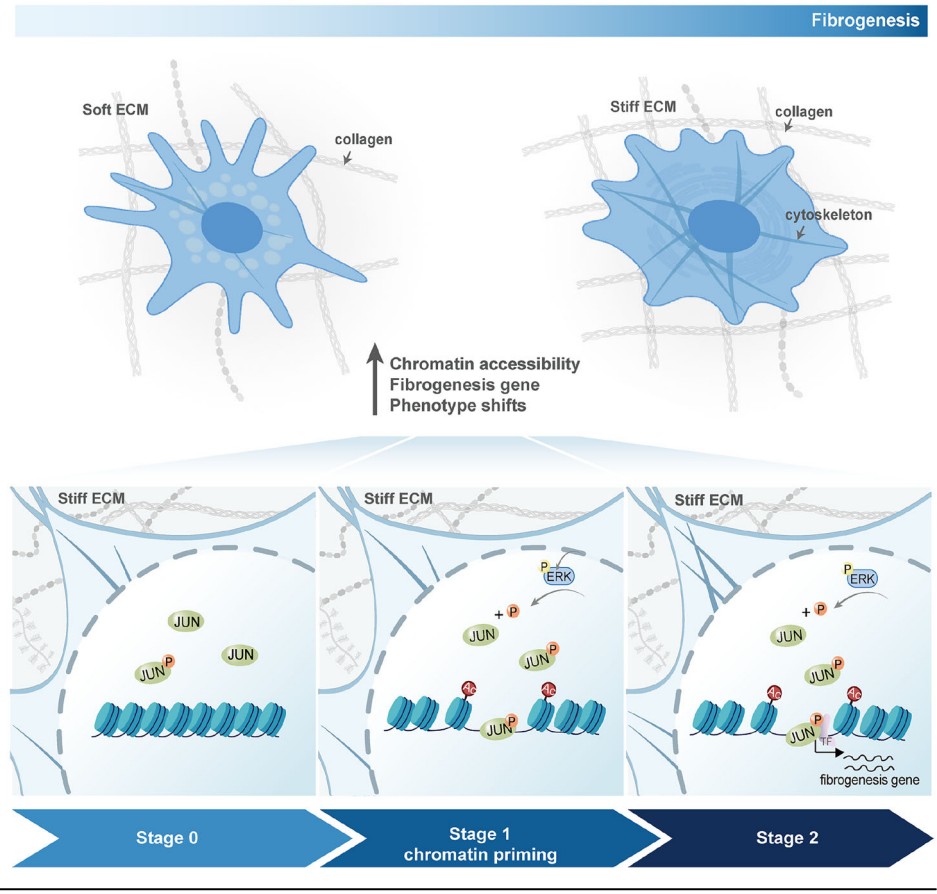

AP-1 family, including the JUN, FOS, and ATF sub-families, which are known as the pioneer TF, can initiate chromatin remodeling and are involved in the regulation of the developmental process[56,88], are involved in the stiffness-induced cell phenotypic transition. Our findings are consistent with previously studied that AP-1 acts as "activating" TF in the activated HSCs by controlling transcription of potent pro-fibrogenic genes[89,90]. Moreover, YAP/TAZ/TEAD and AP-1 form a complex that synergistically activates target genes, and the YAP/TAZ-induced oncogenic growth is strongly enhanced by gain of AP-1 and severely blunted by its loss[91]. Taken together, although there have been several indications that chromatin epigenomic remodeling and TF play a key role in mechanical transduction and cell phenotype shifts, our work reveals a complex interplay by which mechanical signals can be transduced to the nucleus to alter chromatin accessibility through the activation of the regulatory factor JUN, which initiates chromatin priming and drives cell phenotypic shifts. The detailed mechanism between the epigenetic regulators and TFs still needs further study.

In addition, the in vitro system, in which the hydrogel stiffness was adjusted to physiologically relevant conditions of cirrhotic and normal liver tissue, enables us to study the effects of mechanical stimulus on cell behavior during the fibrotic disease process. HSCs are the main contributors of the ECM to the fibrotic liver, increasing the stiffness of the microenvironment. Simultaneously, this stiffness can be detected by HSC surface receptors called integrins, allowing HSC activation and differentiation to myofibroblasts[92,93]. Current models regard the relationship between ECM stiffness and HSC phenotypic transition as a chain of circumstances that results in end-stage fibrotic disease[94]. Although liver cirrhosis is a relatively complex process in vivo, our experiment, built on foundational studies, was able to partially delineate the differentiation process from HSCs to myofibroblasts in response to matrix stiffness. Despite many signaling pathways, molecules, and mechanisms reported to be involved in phenotypic shifts of HSCs[95], it remains unclear to what extent matrix stiffness can drive

phenotypic shifts of HSCs and whether these shifts are accompanied by epigenetic changes. Here, we explored the mechanism underlying the effect of mechanical cues from the microenvironment on cell phenotype using matrix stiffness as the sole variable. More broadly, fibrotic diseases encompass pathological angiogenesis in multiple organs, among which the common feature of fibrosis is conserved.

We reveal that the activated AP-1 could respond to the stiff matrix and establish the primed chromatin state that prepares genes for cell phenotypic shifts. We found that the activated p-JUN plays a key role in transducing the mechanical stimuli to the nucleus. Further studies are needed to explore whether p-ERK nuclear activity sufficient to overcome soft matrix effects, and whether this explicitly requires JUN phosphorylation.

In summary, this work integrates time-resolved, genome-wide profiles to explore fibrotic cell fate and provides insights into the involvement of TF and chromatin dynamics in mechanically related fibrotic diseases and liver homeostasis. Understanding the molecular events involved in the establishment and maintenance of chromatin accessibility and gene regulation in response to mechanical stimulus will benefit the design of strategies for cellular reprogramming, differentiation, and cancer treatment.

## Methods
### Hydrogel formation
The polyacrylamide (PA) hydrogels were prepared following previous protocols[96–98]. Briefly, clean glass coverslips were functionalized using 3-(trimethoxysilyl) propyl methacrylate to facilitate covalent attachment of polyacrylamide gel. The polymer solution contained acrylamide monomers, the crosslinker N, N-methylene-bis-acrylamide, ammonium persulphate, and N, N, N′, N′-tetramethyl-ethylenediamine (TEMED, T22500, Sigma-Aldrich). Polyacrylamide gel was sandwiched between a functionalized coverslip and a dichlorodimethylsilane-treated slide (40140, Sigma-Aldrich). The ratio of % acrylamide/% bis-acrylamide was 7.8/0.8 and 10.4/0.5 for soft and stiff gel, respectively. Subsequently, substrates were

incubated in 0.1 mg/mL N-sulphosuccinimidyl-6-(40-azido-20-nitrophenylamino) hexanoate (sulpho-SANPAH, 22589, Sigma-Aldrich) and activated with ultraviolet light. Finally, a thin layer of plasma fibronectin (0.1 mg/mL, F2006, Sigma-Aldrich) was crosslinked to the gel surface by overnight incubation at 4 °C.

## Cell culture

Immortalized human HSC cell line LX-2 (Procell, CL-0560) was cultured in DMEM supplemented with 10% fetal bovine serum and 1% penicillin-streptomycin at 37 °C with 5% $CO_2$ concentration[20]. Cells were seeded on the hydrogels at 50,000 cells/ml and cultured for 2 or 4 days, and then were collected for downstream experiments.

## Knockdown and overexpression

Mutants and truncations of human *JUN* (Gene ID: 3725) were cloned into the pEGFP-C3 vector for transient expression in LX-2 cells. The JUN-AA mutant featured point mutations where Ser[61] and Ser[62] were changed to Ala, simulating the inhibition of phosphorylation. The JUN-EE mutant contained point mutations where Ser[61] and Ser[62] were altered to Glu, simulating phosphorylation. JUN-EE-Δbasic contained a truncated DNA-binding domain through deletion of the 254–280 gene segment based on JUN-EE. JUN-EE-Δleucine contained a truncated dimerization domain through deletion of the 280–314 gene segment based on JUN-EE. Each of plasmids contains a GFP tag. Cells were transfected with the plasmids using Neofect™ DNA transfection reagent.

## Immunofluorescence

Cells were fixed with 4% paraformaldehyde for 15 min at room temperature, washed twice with PBS, and blocked with 5% fetal bovine serum for 1 h at room temperature. Subsequently, cells were incubated with primary antibodies at room temperature for 1 h, washed three times with PBS, and incubated with the corresponding donkey anti-rat/rabbit Alexa Fluor® 488/594-conjugated secondary antibodies (at final concentration 2 μg/mL) for 1 h at room temperature. The following primary antibodies were used in this study: rabbit anti-α-SMA (Cat. no CY5295, 1:100, Abways), rabbit anti-P-JUN (Cat. 3270 T, 1:100, Cell Signaling Technology). Finally, samples were observed using confocal fluorescence microscopy. The data was processed using ImageJ software. As for the normalized mean fluorescence intensity of immunostaining signals of p-JUN in cells, the fluorescence background in the cytoplasm was subtracted. The total fluorescence intensity was then normalized by dividing the maximum value of each panel to adjust shown range in *y*-axis. Each point on the scatter plot represents a cell. About 15 pairs of cells within the same imaging view were analyzed for each independent experiment.

## Western blotting

Total protein was extracted from the cells using SDS-PAGE protein loading buffer (YEASEN, 20315ES05) containing protease inhibitors (PMSF) and phosphatase inhibitors (Roche, PhosSTOP). Cytoplasmic and nuclear proteins were extracted separately using the Nuclear and Cytoplasmic Protein Extraction Kit (YEASEN, 20126ES50) according to the manufacturer's protocol. Samples were subjected to SDS-PAGE and transferred to nitrocellulose membrane, which was then blocked with 5% skimmed milk for 30 min at room temperature. The membranes were incubated overnight at 4 °C with specific primary antibodies. After washing three times with TBST, the membranes were incubated with secondary antibodies for 2 h at room temperature. Protein signals were acquired using a membrane imaging system (ChemiDoc™ XRS+, Bio-Rad). The following primary antibodies were used in this study: rabbit anti-α-SMA (Cat. no. CY5295, 1:1000, Abways), rabbit anti-p-FOS (Cat. 5348T, 1:1000, Cell Signaling Technology), rabbit anti-P-JUN (Cat. 3270 T, 1:1000, Cell Signaling Technology), rabbit anti-ATF3 (Cat. ab254268, 1:1000, Abcam), mouse anti-c-FOS (Cat. sc-166940, 1:1000, Santa Cruz Biotechnology), mouse anti-c-JUN (Cat. sc-74543, 1:1000, Santa Cruz Biotechnology), mouse-anti-GAPDH (Cat. Ab8245, 1:1000, Abcam). And the secondary antibodies were Goat anti-Rabbit IgG HRP (Cat. BE0101, 1:10,000, EASYBIO), Goat anti-Mouse IgG HRP (Cat. BE0102, 1:10,000, EASYBIO). All the antibodies used in this experiment are listed in Table 1.

All uncropped original blots corresponding to the figures have been uploaded as Supplementary Information, with red boxes indicating the regions used in the manuscript figures. In Supplementary Fig. 3a, the lanes of H3K4me3 and GAPDH shown were derived from the same membrane with different exposures due to distinct molecular weights between target proteins and loading controls. And the lanes of H3K27ac and H3K27me3 were stained and developed separately (due to the membrane size constraints). All quantifications were performed using raw data from the same experimental batch. For experiments conducted over an extended period (Fig. 4d), we used aliquots from the same cell lysate batch for loading control detection in technical repeats, resulting in minor batch variations. In Supplementary Fig. 6d, the uncropped blots were acquired using colorimetric mode, while the main-text figure displays grayscale-converted images processed through ImageJ to align with journal visual guidelines.

Target protein signals were normalized to the corresponding GAPDH signal on each membrane using ImageJ. Biological replicates ($n = 3$ independent experiments) were performed, and representative blots are shown. Experimental and control samples were run in parallel across membranes, with full uncropped blots provided in the Supplementary Information.

## RT-qPCR

TRIzol™ reagent was used to extract total RNA from cells on the matrices according to the manufacturer's instructions. Next, RNA was resuspended in DEPC-treated water, and quantitative RT-PCR was performed using the HiScript® II One Step qRT-PCR SYBR Green Kit according to the product instructions. The Quantagene q225 qPCR System was used to analyze cDNA levels with specific primers and normalize the results to *GAPDH*. The primers used for RT-qPCR are designed by PrimerBank. The primers were validated by examining the primer efficiency and melt curve parameters. The primer efficiency assesses whether the detection of the fluorescent dye reflects this dilution and that the template DNA is truly doubling every cycle. And the primer efficiency falls in the range of 90–110%. Additionally, the melting curve is used to check whether primers are specific to a single gene of interest, and are dissociated at a single temperature. The data shown in this word was passed validation. All qPCR-related primers are shown in supplementary information Table S1.

## Usage of inhibitors

The inhibitors of ERK, JNK, and p38 (Table 1) were diluted to 10 μM as instructed and added to the cells on stiff matrix. After culturing for 2 days, the cells were collected for downstream analysis.

## Fibrosis models

Carbon tetrachloride ($CCl_4$) (ProMab, P07174) liver injury was induced[99,100]. Assign animals to groups indiscriminately using random numbers generated by a computer. The male C57BL/6J (MGI:3028467) mice (weight 24–26 g, 8-week-old, $n = 3$, each group) were injected intraperitoneally twice weekly with an i.p. 1 μL per gram body weight sterile $CCl_4$ in a 1:3 v/v ratio in olive oil or olive oil alone (control) for a duration of 6 weeks. The mice were sacrificed, and their liver tissues were isolated one day after the final injection. Animals were euthanized via cervical dislocation following cervical dislocation.

The mouse livers were isolated and embedded in optimal cutting temperature (OCT) compound for cryo-sectioning. Consecutive liver sections with a thickness of 8 μm were prepared for Masson's staining and IHC experiments. The staining methods were carried out as previously published[96]. The Expression levels of specific molecular markers were determined using immunohistochemistry.

The methods were performed in accordance with relevant guidelines and regulations. All experiments were approved by Institutional Animal Care and Use Committee (IACUC) of Peking University. We have complied with all relevant ethical regulations for animal use. The personnel who

carried out the laboratory technicians were unaware of the specific group assignments of the animals. They conducted the experiments strictly in accordance with the standard operating procedures. The animals were housed in polycarbonate cages with a solid floor. The cages were equipped with nesting material and hiding places. They were group-housed with a maximum of six animals per cage to promote social interaction.

## LC-MS/MS for ECM proteome analysis

After reaching the desired confluence, cells were washed twice with pre-warmed PBS to remove any residual cellular debris or unbound proteins. The culture medium was carefully collected and centrifuged at $1000 \times g$ for 10 min at 4 °C to remove any remaining cell debris. The supernatant was then filtered through a 0.22-μm filter to eliminate any potential bacterial contamination or particulate matter. Proteins in the supernatant were precipitated using an equal volume of ice-cold acetone. The mixture was stored at $-20$ °C for at least 2 h to ensure complete protein precipitation. Following precipitation, samples were centrifuged at $14,000 \times g$ for 20 min at 4 °C. The supernatant was discarded, and the protein pellet was air-dried for 10 min. The protein pellet was resuspended in 50 mM ammonium bicarbonate (ABC) buffer, and the protein concentration was determined using a Bradford assay. Proteins were reduced by adding 5 mM dithiothreitol (DTT) and incubated at 37 °C for 30 min, followed by alkylation with 10 mM iodoacetamide (IAA) in the dark for 15 minutes. Proteins were digested with sequencing-grade trypsin at a ratio of 1:50 (trypsin:protein) at 37 °C for 16 h. The digested peptides were desalted using C18 solid-phase extraction cartridges and dried using a vacuum concentrator. The purified peptide samples were analyzed on the Orbitrap Aatral. MaxQuant and the UniProt Mouse protein database were used to search the acquired spectra. R scripts were used to perform data analysis and visualization.

## RNA-seq library preparation

Total RNA was extracted using the MolPure® Cell RNA Kit (YEASEN, 19231ES50). RNA sequencing libraries were constructed by GENEWIZ, Inc. RNA-seq paired-end reads were sequenced on the Illumina® HiSeq XTen platform.

## ATAC-seq library preparation

The cell pellet was resuspended in moderately cold lysis buffer. The True-Prep DNA Library Prep Kit V2 for Illumina® (TD502) was used for library construction. After 12 cycles of PCR amplification. After purification, paired-end sequencing was performed on the Illumina® Novaseq 6000 platform.

## CUT&Tag library preparation

The CUT&Tag assay was performed using the NovoNGS® CUT&Tag 3.0 High-Sensitivity Kit (NovoProtein, N259-YH01). A total of $1 \times 10^5$ cells were washed twice with 1.5 mL wash buffer and then mixed with activated concanavalin A beads. After successive incubation with the primary (room temperature, 2 h) and secondary (room temperature, 1.5 h) antibodies, cells were washed and incubated with pAG-Tn5 for 1.5 h. The following primary antibodies were used in this study: rabbit anti-H3K27ac (Cat. Ab4729, 1:50, Abcam), rabbit anti-H3K4me3 (Cat. Ab8580, 1:50, Abcam), mouse anti-H3K27me3 (Cat. Ab6002, 1:50, Abcam), rabbit anti-P-FOS (Cat. 5348T, 1:50, Cell Signaling Technology), rabbit anti-P-JUN (Cat. 3270T, 1:50, Cell Signaling Technology), rabbit anti-ATF3 (Cat. ab254268, 1:50, Abcam). And the secondary antibodies were Goat anti-Rabbit IgG HRP (Cat. BE0101, 1:100, EASYBIO), Goat anti-Mouse IgG HRP (Cat. BE0102, 1:100, EASYBIO). Subsequently, tagmentation was performed for 1 h in the provided buffer, and the target DNA fragments were purified using tagmentation DNA extraction beads. After washing the beads in 80% ethanol, the libraries were sequenced using the Illumina® NovaSeq 6000 platform.

## RNA-seq data analysis

The raw sequences were cleaned using TrimGalore[101], and mapped to the human reference genome hg19 by STAR with default parameters. All mapped BAM files were converted to bigwig using bedtools[102] for visualization in IGV. High-quality mapped reads were quantified using htseq-count[103]. Differentially expressed genes were analyzed by DEseq2[104]. Functional enrichment of previously reported gene sets in the transcriptomes was determined using the GSEA software package[105,106], and GO enrichment analysis was performed using DAVID[107]. The gene signatures used in RNA-seq analysis have been reported previously[108]. The active fibrogenic HSC signature was based on the $CCl_4$ model of liver fibrosis, in which differentially expressed genes in HSCs isolated from $CCl_4$-treated mice were compared with HSCs isolated from both healthy and recovered mice. This signature constitutes genes directly associated with ECM deposition and fibrotic HSCs. The conserved initiatory HSC signature was generated by intersecting "in vivo" and "in vitro" genes. The in vivo liver injury signature was defined as up- and downregulated genes in HSCs at 24 h after a single injection of $CCl_4$ as compared with HSCs isolated from healthy mice.

## ATAC-seq data analysis

Sequencing adaptors were removed from the raw ATAC-seq reads using TrimGalore[101], and clean data were mapped to the human reference genome hg19 using Bowtie2[109]. Peaks were called using Macs2[110] with a relaxed q-value threshold of 0.05. The HOMER motif discovery algorithm, *findMotifsGenome.pl*, was used to elucidate the enriched motifs of specific accessible regions[52].

## CUT&Tag data analysis

TrimGalore[101] was used to cut adapters, and trimmed reads were then aligned to the human genome hg19 using Bowtie2[109]. Reads were sorted and converted to BAM format, and duplicates were marked using Picard. Chromatin markers were used to train the chromatin states in mouse HSCs by chromHMM[42]. For comparison of the ChIP-seq and ENCODE data, H3K4me1, H3K27ac, and CTCF ChIP-seq data generated in MEFs were downloaded from ENCODE database. After normalization of the sequencing depth, the deepTools2 software with default parameters[111] was used to plot the heatmaps showing signals around peak regions.

## Identification of unique and shared differentially accessible regions

The read pair numbers inside each ATAC-seq peak were calculated using DiffBind[112]. The differentially accessible regions were identified using DEseq2[104] with a cutoff of Fold Change $\geq 2$ and $p$ value $< 0.05$ through pairwise spatiotemporal comparisons (D2 soft vs. D2 stiff, D4 soft vs. D4 stiff, D2 soft vs. D4 soft, D4 stiff vs. D4 stiff). Four unique clusters and four shared clusters were selected from the unique and shared peaks; for example, D4 stiff unique peaks were selected from the differentially accessible regions with higher signals by intersecting D2 stiff vs. D4 stiff and D4 soft vs. D4 stiff.

## Statistics and Reproducibility

The high-through sequencing experiments consisted of at least two biological replicates. And the biochemical assays consisted of at least three biological replicates. For comparison between conditions in this study, two-tailed Wilcoxon tests, paired Wilcoxon tests, or one-way ANOVA were performed using the function in R. The center line of a boxplot represents the median. The data expressed as the mean ± standard deviation (SD) was representative of all independent experiments. $P$ values of $< 0.05$ were considered significant.

## Reporting summary

Further information on research design is available in the Nature Portfolio Reporting Summary linked to this article.

## Data availability

The raw data of ATAC-seq, RNA-seq, and CUT&Tag have been deposited at Gene Expression Omnibus (GEO) under the accession number GSE220703. The uncropped and unedited blot images were included in the

Supplementary Information. All the raw proteomics data were deposited at iProX under the project ID IPX0011739000. The numerical source data for graphs and charts is shown in Supplementary Data 1.

## Code availability

The in-house written R-scripts required to reanalyze the data reported in this paper is available from the lead contact upon request.

## Materials availability

This study did not generate new, unique reagents and biological materials.

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

## Acknowledgements

W.Y., T.D. and Y.S. were supported by the National Key R&D Program of China (No. 2022YFA3401100) and the National Science Foundation of China (22127804). W.Z. and C.L. were supported by the National Natural Science Foundation of China (32288102, 32025006) and the National Key Research and Development Program of China (2021YFA1100300). Z.F. was supported by the S&T Program of Hebei No.22377703D. W.Q. was supported by Hebei Natural Science Foundation (Project H2021206314). Part of the data analysis was performed on the High-Performance Computing Platform of the Center for Life Sciences, Peking University. The secretome proteomics experiments were supported by Dr. Zhang Xiaohui and Dr. Shi Xiaoming from the State Key Laboratory of Natural and Biomimetic Drugs at Peking University.

## Author contributions

Y.S. and C.L. supervised the study. Z.F. and W.Q. gave some of the experimental guidance and provided part of samples. W.Z. and W.Y. designed the experiments and completed the sequencing library construction. W.Y. and T.D. conducted all the imaging experiments, chemical assays and data analysis. W.Z. performed the library construction and bioinformatics analysis. W.Z. W.Y. and T.D. wrote the manuscript with input from all authors.

## Competing interests

The authors declare no competing interests.
