## [Transparent Peer Review file · Communications Biology]

Increased matrix stiffness promotes fibrogenesis of hepatic stellate cells through AP-1-induced chromatin priming

Corresponding Author: Dr Wenxue Zhao

Version 0:

Reviewer comments:

Reviewer #1

(Remarks to the Author)

Major comments:

General

A major issue of the paper is that the cells used are a cell line which has been serially passaged on plastic. If the premise of the paper is that matrix rigidity alters fibrogenic chromatin accessibility, the cells are theoretically already in this fibrogenic primed state, and what the authors observe in their hydrogels is actually a “reversibility” of this priming. How can the authors resolve this issue? At minimum, the ATAC/RNA profile of these cells on plastic should be analyzed and compared to their hydrogel analysis. If possible, would greatly benefit the paper to use freshly isolated cells from the tissue and use these fresh cells in their hydrogel system and at least perform qPCR analysis to show correlation.

The primary focus of the paper is fibrosis, but no fibrosis assays are provided other than just qPCR for ACTA2. The authors need to provide fibrogenic assays such as ECM deposition, matrix remodeling, cell contractility, etc. to flush out the mechanisms provided here.

I'm failing to observe significant novelty in this project since matrix stiffness induced chromatin changes have already been examined within the context of fibrosis in addition to AP1/JUN/FOS (discussed below). However, the plasmids used in Figure 5 offer potential to provide unique mechanistic insight into these phenomena. I suggest the authors use these materials to their advantage (discussed below).

Minor comments:

General

Several papers have already examined the effect of matrix stiffness on cells using RNAseq/ATACseq within the context of fibrosis...these papers should be referenced and preferentially discussed (PMIDs – 33875841, 33625469, 35996026, 37216538, 37727069, 36176599). In addition, papers examining AP1 within the context of fibrosis should also be referenced (example PMIDs - 28424250).

In general, it would elevate readability if the authors reduced figures to show just the message of the figure panel. I appreciate providing all data for reviewers/readers to assess the work, but it would be helpful to remove not necessary data into supplement. An example is Figure 4f, do all 8 groups need to be shown here?

Figure 1

Fig 1b (and Fig. S1a) – protein analysis of SMA should be analyzed in addition to the Acta2 qPCR data provided in Fig. S1a

Figure 2

-the data are represented by discussing day 4 data first, then day 2 data. This makes the reader believe that these data were collected within different experiments. If true, than these two independent datasets can't be compared to each other. If they were collected within the same experiment, than it would be more clear to present the day 2 data first, than present the day 4 data.

-What is the overlap of differential ATAC sites at day 2 vs day 4? For example, are all the increased ATAC sites at day 2 also increased at day 4? Likewise for the decreased ATAC sites.

-the nomenclature for the gene groupings is confusing and needs to be clarified in the results section

-since the GO terms aren't followed up at all in the manuscript, they are entirely descriptive. Would make the figure more clear to put these into supplement.

Figure 3

The heatmap in figure 3a needs to be more clear. In particular, it's not clear what the list on the left is supposed to represent. What does the color scale on the right heatmaps represent? In the most right-hand heatmap, all the samples look to be similar and there are no differences?

Figure 3d – this should be reproduced and quantified using western blots

Figure 5

Can the authors provide a simple venn diagram of the overlap between the matrix stiffness induced ATAC changes with the phospho-JUN induced peaks?

It would help this figure to use these JUN loss of function cells and show a loss of transcriptional changes (using their original RNAseq data as guidance) upon seeding the cells in hydrogels. For example, do they no longer see a reduction in MMP gene expression in stiff matrices? Do they no longer see an elevation of COL8A1?

Is it known whether or not phospho-JUN is elevated in mesenchymal cells in liver fibrosis using tissue sections? If not, can the authors test this? Would greatly strengthen this figure.

Reviewer #2

(Remarks to the Author)

This potentially useful manuscript will add significantly to the literature on the molecular basis of hepatic and other forms of fibrosis in response to matrix stiffness via chromatin remodeling. The authors focus on a single hepatic stellate cell model responding to two substrates of differing stiffness, i.e., polyacrylamide gels with differing compliance coated with fibronectin, after two or four days. They show original, interesting patterns of altered gene expression and chromatin priming using ATAC-seq, Cut&Tag, and histone modification characterization to identify apparent roles for ERK as well as JUN as part of an AP-1 transcription factor. The results presented are of immediate relevance to researchers on hepatic fibrosis and somewhat to those in the area of mechanotransduction. In general, the conclusions appear reasonable with good analyses of JUN function, but with the following major concern #1 below.

1. The authors very puzzlingly focus on ITGB2 (integrin subunit beta 2) as their main marker of fibrosis in cells. Their reference to use of this putative marker (Berzigotti, 2017) does not provide adequate justification at all for this choice, and this particular marker is not a generally accepted marker of fibrosis. For hepatic fibrosis and other types of fibrosis, the usual markers are collagen (types I, III, IV, etc.), fibronectin, and sometimes laminin, MMPs, and TIMPs, with CTGF for other cell types, as well as other markers. In fact, integrin markers for liver fibrosis are to my knowledge not the integrin beta2 subunit, but instead alphaV-beta6 and other alphaV subunit integrins. At a minimum, the authors need to evaluate the expression at days 2 and 4 of at least some of these classical markers of fibrogenesis. Their use of versican can help slightly, but it is not as well-characterized as the other markers. This issue is critical.

2. The authors focus on JUN and ERK as key regulators, with good evidence for a role for JUN (in an assumed AP-1 complex) and inhibitor evidence for a role for prolonged activation of ERK. However, this conclusion seems over-simplified because the authors have not shown that simple experimental elevation of p-ERK by some stimulus other than stiffness will elevate p-JUN and lead to chromatin remodeling. The authors seem to acknowledge this weakness, but it appears to be quite significant and weakens their conclusions. At a minimum, if no experimental support can be provided, they would need to emphasize that their findings are descriptive and correlative because inhibitor studies can only identify a requirement for some process, not causality.

Less major:

3. The authors are in a chicken-versus-egg situation in that matrix stiffness itself is the result of transcriptional changes in matrix secretion, so it would help if they would clarify why they are studying only cell responses to stiffness with respect to fibrogenesis. Most likely, they are examining part of a positive feedback cycle of cell activation, matrix synthesis and assembly to increase stiffness, which in turn induces their types of response with more matrix synthesis.

4. The ITGB2 marker they use is often a marker for certain immune cells, and it is puzzling that they also identify a major gene ontology category of "antigen processing and presentation" by their ATAC-seq and RNA-seq analyses. How can this strange GO category be explained for an HSC line?

5. Minor typos on line 457: "could response to the stiffen matrix"

Version 1:

Reviewer comments:

Reviewer #1

(Remarks to the Author)

I thank the authors for addressing my concerns and comments. The newly generated data and discussions greatly strengthen the claims in the manuscript.

My last remaining suggestion is to include the additional newly generated data & figures located in the rebuttal letter (eg Figure R4) into the manuscript, either in the main figures or supplement.

Also, the proteomics data of cells on soft vs stiff matrices is powerful data and quite interesting and should go in the main figures.

Reviewer #2

(Remarks to the Author)

The authors have responded fully and conscientiously to my own comments. I support acceptance for publication if the other reviewer is also sufficiently positive.

Very minor point:

In lines 468-470, there are two very minor grammatical issues (play vs. plays and probably a period instead of the comma):
“We found that the activated p-JUN play a key role in transducing the mechanical stimuli to the nucleus, further studies are needed to explore whether p-ERK nuclear activity sufficient to overcome soft matrix effects”

Version 2:

Reviewer comments:

Reviewer #1

(Remarks to the Author)

I thank the authors for completely addressing all of my comments. I recommend this manuscript for publication.

Point-to-point Response

We appreciate the valuable comments and suggestions from the reviewers. We have used them as a springboard to improve our manuscript by performing new experiments and analyses. Below is the point-to-point response to the comments. Please note that all modifications in the manuscript are highlighted with red for your easy reference. With these revisions, new data and analyses, and clarifications, we hope that the reviewers will find this re-submitted manuscript sufficiently improved, more cogently and convincingly presented, and acceptable for publication.

Reviewer #1

Major comments:

General

A major issue of the paper is that the cells used are a cell line which has been serially passaged on plastic. If the premise of the paper is that matrix rigidity alters fibrogenic chromatin accessibility, the cells are theoretically already in this fibrogenic primed state, and what the authors observe in their hydrogels is actually a “reversibility” of this priming. How can the authors resolve this issue? At minimum, the ATAC/RNA profile of these cells on plastic should be analyzed and compared to their hydrogel analysis. If possible, would greatly benefit the paper to use freshly isolated cells from the tissue and use these fresh cells in their hydrogel system and at least perform qPCR analysis to show correlation.

Response: Thanks for the insightful comments and for raising a critical point regarding the use of the LX-2 cell line in our study. In response to your concern, we have provided additional data and evidence regarding the experimental design and revised the main text lines 106–117 to illustrate our biological model.

The primary human hepatic stellate cells (HSCs) are known to undergo myofibroblastic transdifferentiation during the process of liver fibrosis, that is, transforming from the quiescent state to the active state. The LX-2 cell line is a commonly used cell line for studying liver fibrosis/cirrhosis since the access to primary human HSCs is limited. When LX-2 cells are passaged on tissue culture plastic (TCP), they may be subjected to stiffness stimulation at the GPa scale. To assess their baseline state before being subjected to the hydrogel matrices and subsequent changes, we measured the expression changes of active HSCs marker genes (*ACTA2*, *COL1A1*, *COL3A1*, ...) by qPCR of LX-2 cells at several critical timing points, including

cells on TCP, resuspension, reseeding, and culturing on matrices for several days (Fig.R1a, b). The results showed that when cells were transferred from TCP to a matrix, they indeed underwent a ‘reversibility’ process with the gene expression patterns reset to quiescent HSC state within several hours to one day, which is likely in accordance to the process of cells being trypsin digestion, resuspension, reseeded on the matrices, and re-adherence. Subsequently, the cells were observed to gradually establish a new steady state on either a stiff or soft matrix during the next 2 to 3 days. Based on these results, we collected cells cultured for 2 days and 4 days on matrices to study their mechanoresponsive mechanisms (Fig.R1c).

Fig. R1 (a) Schematic of cell phenotypic transition process when being isolated and reseeded onto matrices. **(b)** The relative expression levels of fibrotic genes in LX-2 cell lines over a period of days. The cells were initially plated on TCP and then isolated and reseeded onto soft and stiff matrices and TCP (used as a control). The results are shown as mean \pm standard deviation. $n = 3$ per group per time point. **(c)** An overview of the study design depicting the workflow for cell culture, multi-omics analysis and experimental assays.

Our results are also supported by the previous studies^{1,2}. In Caliri and his colleagues' research, the human primary HSCs were mechanically primed on TCP for 7 days and then transferred to either a soft or stiff matrix for further cultivation (Fig.R2a). They observed that the cells on soft hydrogels quickly adjusted to the new mechanical environment and display the phenotype like quiescent HSCs (light blue line in Fig.R2b, c). The cells on stiff hydrogels showed increased spreading within the first 5 days and then maintained the myofibroblast-like morphology, rather than a continuous reverse due to the transfer from the TCP environment to the matrix (dark blue line in Fig.R2b, c). These results suggested that the reversible process of cells mainly takes place during digestion, resuspension and reseeded of the cells. The data also indicated that quickly after being transferred from TCP to matrices, cells start to set on myofibroblastic transdifferentiation and eventually adapt to the new mechanical environment.

Fig.R2 This figure is from Steven *et. al*, with a red box added to highlight our point. (a) Schematic of experimental design for stellate cell culture on hydrogels. Stellate cells were initially plated on TCPS or directly onto soft static hydrogels following isolation. After

7 days of mechanical priming, cells were moved to either soft or stiff hydrogels or onto stiff-to-soft hydrogels. As a control, freshly isolated cells that were plated onto soft hydrogels were cultured for an additional 14 days (21 days total) with no mechanical priming. **(b)** Stellate cell spread area after plating on hydrogels and culturing for an additional 14 days. *******, $p < 0.001$. **(c)** Stellate cell phase contrast images plating on hydrogels and culturing for an additional 14 days. Scale bars, 100 μm . $n > 37$ cells per group per time point.

Moreover, previous evidence showed that chromatin remodeling would also adapt in a rapid fashion to soft basal levels after being transferred from stiff substrates². Killaars et al. studied hMSCs cultured on stiff substrates ($E = 32.7$ kPa) which was subsequently softened *in situ* with light to a final $E = 5.5$ kPa. They assessed the chromatin remodeling by quantification of histone acetylation levels, nuclear volume, and chromatin condensation parameter (CCP) at multiple time points and found that the chromatin remodeling was rapid, ranging from 0.5 h to 1 day (Killaars et al, 2019, Figure 4)². To further address whether the matrix stiffness-induced chromatin accessibility is a “reversibility” of this mechanical priming on TCP or responding to the new stiff matrix, we compared the accessibility patterns of primed chromatin regions defined in our study (main text, Fig.2) among cells on TCP, soft or stiff matrices, and primary HSCs. The results showed that the primary HSCs display chromatin states similar to those observed on soft matrix (Fig.R3). Importantly, cells on the stiff matrix for 2 days and 4 days exhibit higher accessibility in primed chromatin than those on TCP, possibly because cells on TCP have already reached a long-term homeostasis state, while those on the stiff matrix for 2 days and 4 days are in the process of responding to stiffness and being activated. Therefore, the observed chromatin priming phenomenon was attributed to matrix stiffness.

Fig.R3 A heatmap of primed chromatin regions between the different culture conditions and primary HSCs, demonstrating the similarity in accessibility with cultures from soft matrices (“D2” means cultured for two days, “D4” means cultured for four days). Each row

represents a primed chromatin region. Each column represents different culture conditions, and the hierarchical clustering results show the similarity between samples.

Next, we examined the implications of our results for the use of soft versus stiff matrices for recapitulation of the *in vivo* quiescent and active HSCs. By comparing with the gene expression of HSCs from cirrhotic patients vs. healthy donor, cells on the soft and stiff matrices were respectively enriched with the signatures of quiescent and active HSCs, indicating their potential in respectively stimulating the gene expression patterns of *in vivo* quiescent or active HSCs (Fig.R4a, b). Although the LX-2 cell line is typically cultured on TCP, with optimized experimental design, it is feasible to use stiff and soft substrates to recapitulate pathological and physiological conditions, enabling exploration of cell phenotypic transitions and underlying mechanisms during disease progression.

Fig.R4 (a–b) The GSEA results show that cells cultured on soft and stiff matrices for 2 days **(a)** and 4 days **(b)** were enriched with marker genes of quiescent and active HSCs respectively.

Finally, investigating the reversibility of HSC cells may also be a very interesting research direction. A more suitable experimental design might involve using a system with variable stiffness, allowing the cellular mechano-environment to change *in situ* without the need to re-digest and re-seed the cells, as described by some previous studies³⁻⁵.

The primary focus of the paper is fibrosis, but no fibrosis assays are provided other than just qPCR for ACTA2. The authors need to provide fibrogenic assays such as ECM deposition, matrix remodeling, cell contractility, etc. to flush out the mechanisms provided here.

Response: We sincerely appreciate the reviewer's valuable comments and suggestions. To quantify the fibrotic phenotypes for cells cultured on soft and stiff matrices, we performed the secretory proteomic approach using liquid

chromatography-tandem mass spectrometry (LC-MS/MS) to analyze the ECM deposition of cells on different matrices over periods of 2 and 4 days (aligning with the time points of other omics data collected in our study). The results revealed that the expression of core ECM proteins in cells cultured on the stiff matrix was generally upregulated compared to those cultured on soft substrates (Fig.R5a–d). The core ECM proteins, including collagens, ECM glycoproteins and proteoglycans, assemble and remodel ECM and regulate cellular functions. Compositions of ECM are qualitatively and quantitatively dynamic during tissue fibrosis⁶. This finding is consistent with the previous research, in which Wu et al., using LC-MS/MS, analyzed and found that the expression of core ECM proteins in the fibrotic livers of CCl₄-treated mice was upregulated⁷. In our study, the increased expression of core ECM proteins between cells cultured on matrices of varying stiffness indicates that the stiff matrix induces cell activation and increases the degree of fibrosis.

Fig.R5 (a, c) The proportion of core ECM proteins that are upregulated in cells cultured on a stiff matrix for 2 days **(a)** and 4 days **(c)**, relative to the total number of proteins detected. **(b, d)** The expression changes of the three categories of core matrix proteins detected in cells cultured for 2 days **(b)** and 4 days **(d)** on stiff vs. soft matrices.

We have also conducted traction force microscopy (TFM) experiments to measure cell contractility. However, we encountered a technical challenge where the high stiffness of the matrix (40 kPa) appears to limit the movement of fluorescent tracer particles, which is crucial for assessing cell contractility. This makes it difficult to

compare the results with those obtained on the soft matrix (2 kPa). The results showed that cellular traction force exerted by cells on the soft matrix was around 80 Pa (Fig.R5a, b), which is consistent with the previous study⁸. Wang et al. measured the traction force of the quiescent HSCs to be 12 nN, which, when converted based on the cell area, is also roughly around 100 Pa. They also measured the traction force of active HSCs which is 100 nN⁸. While the contact area between the cell and the stiff matrix is about ten times larger than that with the soft matrix, the contraction force per unit area is not significantly different from that of cells on a soft matrix.

Fig.R5 (a) Traction force microscopy (TFM) using the polyacrylamide gel with fluorescence beads as the matrix. TFM converts cell adhesive force to matrix deformation. **(b)** The representative images and traction force of cell on the soft matrix from TFM experiments. The bright-field image shows the cell. Color spectrum indicates stress magnitude (Pa), with areas of low traction in blue and high traction in red. Scale bar, 20 μm.

By comparing with other cell lines such as cardiomyocytes and epithelial cells^{9,10}, the contractile force of the hepatic stellate cell line LX-2 is relatively low. Moreover, in our research system, the stiff substrate imposes limitations on the movement of beads in the TFM experiments. Nevertheless, TFM still provides a general range for the cellular contractile force, as pulling beads in a 40 kPa matrix is estimated to require a force of approximately 100 Pa, consistent with previous research findings^{10,11}.

I'm failing to observe significant novelty in this project since matrix stiffness induced chromatin changes have already been examined within the context of fibrosis in addition to AP1/JUN/FOS (discussed below). However, the plasmids used in Figure 5 offer potential to provide unique mechanistic insight into these phenomena. I suggest the authors use these materials to their advantage (discussed below).

Response: We thank the reviewer for the insightful suggestions. . We have expanded our analysis and included additional data that leverage the plasmids used in our experiments to provide a more in-depth and unique mechanistic insight into the phenomena under investigation (Figure 5 in the revised manuscript). We believe that

these revisions will not only address the concerns about novelty but also strengthen the overall contribution of our work to the field.

Minor comments:

General

Several papers have already examined the effect of matrix stiffness on cells using RNAseq/ATACseq within the context of fibrosis. These papers should be referenced and preferentially discussed (PMIDs – 33875841, 33625469, 35996026, 37216538, 37727069, 36176599). In addition, papers examining AP1 within the context of fibrosis should also be referenced (example PMIDs - 28424250).

Response: We thank the reviewer for pointing out the importance of citing relevant literature in the context of our study. We have added these references to the main text and provided a detailed discussion on how our findings are related and advance the understanding presented in these studies.

Evaluation of genome-wide chromatin accessibility using ATAC-seq of cells stimulated by mechanical cues (including stiffness, stretch and topography) revealed a dramatic effect on the global chromatin structure^{4,12-16}. However, the chromatin structure of cells does not always follow similar trends in chromatin accessibility in response to microenvironmental stiffness cues. For instance, one previous study found that mesenchymal stem cells (MSCs) maintain more accessible chromatin status on soft substrates than those on stiff substrates¹⁵, while other works reported that stiff matrix resulted in more accessible chromatin of fibroblast² and mammary epithelial cells¹⁶. Our results showed that stiff matrix induced the establishment of primed chromatin in hepatic stellate cells (HSCs), in which the chromatin regions become more accessible prior to the upregulation of nearby genes. Since HSCs have also been described as a liver-resident MSCs population and can transdifferentiate to myofibroblasts^{17,18}, it is interesting to find that these cell types have distinct mechanisms regulating chromatin accessibility upon mechanical stimulus, indicating that there are still many issues worth studying between cell fate and the response to mechanical signals.

Moreover, previous studies have reported that *JUN* gene expression is increased in human fibrotic diseases and that systemic induction of c-Jun in mice resulted in development of fibrosis of multiple organs^{19,20}. The upregulation of transcription factors

or their phosphorylation and activation both have an impact on their transcriptional activation capacity²¹. Our research found and demonstrated that the transcription factor JUN did not show significant changes at the gene and protein expression levels; instead, its phosphorylation and activation status responded to mechanical signals. Additionally, scRNA-seq data from patients with liver cirrhosis also revealed no significant changes in *JUN* gene expression. Our results indicated the impact of the phosphorylation and activation of transcription factors on their transcriptional activity.

In general, it would elevate readability if the authors reduced figures to show just the message of the figure panel. I appreciate providing all data for reviewers/readers to assess the work, but it would be helpful to remove not necessary data into supplement. An example is Figure 4f, do all 8 groups need to be shown here?

Response: We have revised the main figure according to the reviewer's suggestion, moving the descriptive results (GO terms) to the supplementary material. Additionally, we have modified Figure 4f to display only the primed chromatin group.

Figure 1

Fig 1b (and Fig.S1a)-protein analysis of SMA should be analyzed in addition to the Acta2 qPCR data provided in Fig. S1a

Response: We performed the α -SMA western blot assay in addition to the qPCR data, and the results showed that a slightly up-regulated after 2 days of culture on a stiff matrix, and more significantly up-regulated after 4 days of culture (Fig.R6a, b), which were consistent with the transcription patterns (main text Fig.S1).

Fig.R6 (a–b) Western blot images showing the protein expression levels of α -SMA in cells cultured for 2 days (**a**) and 4 days (**b**) on the soft and stiff matrices. GAPDH served as the loading control.

Figure 2

-the data are represented by discussing day 4 data first, then day 2 data. This makes the reader believe that these data were collected within different experiments. If true,

than these two independent datasets can't be compared to each other. If they were collected within the same experiment, then it would be more clear to present the day 2 data first, than present the day 4 data.

Response: These data shown in Figure 2 were collected within the same experiment. We have made revisions to Fig.2a-b and the corresponding content in the manuscript, presenting the day 2 data first, and then presenting the day 4 data.

-What is the overlap of differential ATAC sites at day 2 vs day 4? For example, are all the increased ATAC sites at day 2 also increased at day 4? Likewise for the decreased ATAC sites.

Response: To address the question, we have included the relevant analysis in our manuscript, specifically in Fig.2c of the main text. As shown in this figure, there is a significant overlap in the differential ATAC-seq sites between these two time points. There are 1983 overlap ATAC-seq sites are increased in cells on the stiff at day 2 vs day 4, and 348 overlap ATAC-seq sites are decreased.

Perhaps our nomenclature in this figure has caused some confusion. The first four groups of this heatmap represent regions that are specifically increased ATAC-seq sites under different culture conditions. For example, "D2 soft" represents the ATAC-seq sites that are specifically increased in cells cultured on the soft matrix for two days (Fig.R7d), and "D2 stiff" represents the ATAC-seq sites that are specifically increased in cells cultured on the stiff matrix for two days (Fig.R7a). While the last four groups of this heatmap represent sites that are commonly increased under two of the culture conditions. For example, "D2 soft + D4 soft" represents the ATAC-seq sites that are commonly decreased both in cells cultured on the stiff matrix for two and four days (Fig.R7f). "D2 stiff + D4 stiff" represents the ATAC-seq sites that are commonly increased both in cells cultured on the stiff matrix for two and four days (Fig.R7c).

Fig.R7 Heatmap shows ATAC-seq signals for unique and shared peak groups and adjacent gene RNA-seq patterns for echo groups (“D2” means cultured for two days, “D4” means cultured for four days). Each row in the heatmap represents one ATAC-seq site. **(a, d)** The ATAC-seq sites that are specifically increased **(a)** or decreased **(d)** in cells cultured on the stiff matrix for two days. **(b, e)** The ATAC-seq sites that are specifically increased **(b)** or decreased **(e)** in cells cultured on the stiff matrix for four days. **(c, f)** The overlap ATAC-seq sites that are commonly increased **(c)** or decreased **(f)** both in cells cultured on the stiff matrix for two and four days.

-the nomenclature for the gene groupings is confusing and needs to be clarified in the results section.

Response: Thanks for the insightful comments, we have clarified the nomenclature for the gene groupings in figure 2 and the main text lines 176–181 (also refer to Fig.R7).

-since the GO terms aren’t followed up at all in the manuscript, they are entirely descriptive. Would make the figure more clear to put these into supplement.

Response: We thank the reviewer for the suggestion. We have moved the GO terms results (Fig.1e, f) to the supplementary material to enhance the clarity of the main figure.

Figure 3

The heatmap in figure 3a needs to be more clear. In particular, it's not clear what the list on the left is supposed to represent. What does the color scale on the right heatmaps represent? In the most right-hand heatmap, all the samples look to be similar and there are no differences?

Response: In Fig.3a left panel, ten epigenomic-marked chromatin states of hepatic stellate cells was classified by the chromHMM, using a multivariate hidden Markov model ²². The heatmaps showed candidate-state descriptions for each state. The different shades of the same color were used to distinguish the chromatin state.

The middle section of Fig.3a displays a heatmap of the emission parameters in which each row corresponds to a different state, and each column corresponds to a different mark defined on the basis of the ChIP-seq for three histone modifications (H3K27me3, H3K27ac, and H3K4me3), ATAC-seq and RNA-seq. The darker blue color corresponds to a greater probability of observing the mark in the state.

In Fig.3a right panel, the heatmap displays the fold enrichment for the eight chromatin accessibility clusters in epigenomic-marked chromatin states (annotated in left panel). Similarly, a darker blue color corresponds to a greater fold enrichment, and there is one color scale for the entire heatmap. Compared to other clusters, the primed chromatin specifically shows enrichment in both active enhancers and bivalent promoters (which were highlighted by red box). While some other clusters exhibit similar patterns, we believe that the primed chromatin has a more pronounced pattern, and our current work was focused on the primed region. According to reviewer's suggestion, we have added more explanations in the figure legend.

Figure 3d – this should be reproduced and quantified using western blots.

Response: Thanks for the reviewer's valuable suggestion. We have reproduced the IF image and revised Figure 3d in the main text. We quantified the relative change of histone modification using western blots, and the results showed that there are no significant changes in the levels of these histone modifications (Fig.R8a, b). These results are consistent with our main findings and indicate that the specific increased H3K27ac modification (examined by CUT&Tag) in primed chromatin regions may result in the chromatin priming (main text Fig.3b).

Fig.R8 (a) Representative immunofluorescence images of cells cultured on different matrices for two days. Anti-H3K27ac (yellow), anti-H3K4me3 (red), and anti-H3K27me3 (green). scale bar, 10 μ m. **(b)** Western blots showing the relative change of histone modification. GAPDH served as the loading control.

Figure 5

Can the authors provide a simple venn diagram of the overlap between the matrix stiffness induced ATAC changes with the phospho-JUN induced peaks?

Response: We have provided a venn diagram to illustrate the overlap between the matrix stiffness-induced ATAC-seq changes (primed chromatin regions) and the phospho-JUN induced peaks. The results indicate that the overlap is approximately a quarter of the total changes (Fig.R9).

Fig.R9 The venn diagram showing the overlap between the matrix stiffness induced ATAC-seq changes with the phospho-JUN induced peaks.

It would help this figure to use these JUN loss of function cells and show a loss of transcriptional changes (using their original RNA-seq data as guidance) upon seeding the cells in hydrogels. For example, do they no longer see a reduction in MMP gene expression in stiff matrices? Do they no longer see an elevation of COL8A1?

Response: Thanks for the reviewer's valuable suggestions. We have examined the transcriptional changes of *MMP9*, *MMP2* (both showing reduction in the original RNA-seq data), and α -*SMA*, *COL8A1* (both showing increase in the original RNA-seq data) upon the loss of function of p-JUN. The results showed that the expression of *MMP9* was increased, while the expression of *COL8A1* was decreased in cells with deficient p-JUN on the stiff matrix (Fig. R10).

Fig. R10 Relative mRNA expression of *MMP9*, *MMP2*, α -*SMA* and *COL8A1* in cells had JUN loss of function on the stiff matrix. The expression level was normalized with *GAPDH*, 3 independent experiments.

Is it known whether or not phospho-JUN is elevated in mesenchymal cells in liver fibrosis using tissue sections? If not, can the authors test this? Would greatly strengthen this figure.

Response: Thanks for reviewers' insightful suggestions. Upon reviewing the current publication, it appears that the elevation of phospho-JUN in mesenchymal cells, specifically hepatic stellate cells, during liver fibrosis has not been conclusively established. To address this gap in knowledge and to strengthen our findings, we have conducted additional experiments using the CCl₄-treated fibrotic mouse model. The chronic fibrosis mouse model was established following methods from the previous publication, involving the intraperitoneal injection of CCl₄ for 6 weeks, which is a widely used and reproducible model for chronic fibrosis that can replicate many characteristics of human fibrotic liver disease (Fig.R11a, b). On consecutive sections of fibrotic liver and control liver, activated hepatic stellate cells were marked with α -SMA antibody, and the p-JUN was also labeled. The results showed an increase in p-JUN protein expression in hepatic stellate cells after fibrosis (Fig.R12a, b). This indicates that the phosphorylation and activation of the transcription factor JUN may

play an important role in the fibrotic process *in vivo*. We have added these results to the main text lines 334–341 and Fig.5.

Fig.R11 Mouse model of liver fibrosis induced by CCl₄ administration. **(a)** CCl₄ treatment schedule. **(b)** Visualization of the degree of liver fibrosis by Masson's trichrome staining, where collagen fibers are stained blue. Scale bar, 200 μ m.

Fig.R12 IHC characterizes the expression levels of p-JUN in fibrotic mouse liver cells. **(a–b)** Consecutive sections from fibrotic liver **(a)** and control liver **(b)** are stained with α -SMA antibody to mark activated hepatic stellate cells, and with the p-JUN. Scale bar, 200 μ m. Enlarged views of the areas enclosed by the dashed boxes are shown below, scale bar, 100 μ m. The black arrows in the images emphasize the regions where hepatic stellate cells are located.

Reviewer #2

This potentially useful manuscript will add significantly to the literature on the molecular basis of hepatic and other forms of fibrosis in response to matrix stiffness via chromatin remodeling. The authors focus on a single hepatic stellate cell model responding to two substrates of differing stiffness, i.e., polyacrylamide gels with differing They show original, interesting patterns of altered gene expression and chromatin priming using ATAC-seq, Cut&Tag, and histone modification characterization to identify apparent roles for ERK as well as JUN as part of an AP-1 transcription factor. The results presented are of immediate relevance to researchers on hepatic fibrosis and somewhat to those in the area of mechanotransduction. In general, the conclusions appear reasonable with good analyses of JUN function, but with the following major concern #1 below.

1. The authors very puzzlingly focus on ITGB2 (integrin subunit beta 2) as their main marker of fibrosis in cells. Their reference to use of this putative marker (Berzigotti, 2017) does not provide adequate justification at all for this choice, and this particular marker is not a generally accepted marker of fibrosis. For hepatic fibrosis and other types of fibrosis, the usual markers are collagen (types I, III, IV, etc.), fibronectin, and sometimes laminin, MMPs, and TIMPs, with CTGF for other cell types, as well as other markers. In fact, integrin markers for liver fibrosis are to my knowledge not the integrin beta2 subunit, but instead alphaV-beta6 and other alphaV subunit integrins. At a minimum, the authors need to evaluate the expression at days 2 and 4 of at least some of these classical markers of fibrogenesis. Their use of versican can help slightly, but it is not as well-characterized as the other markers. This issue is critical.

Response: We thank the reviewer for the positive comments and constructive suggestions. Following the reviewer's suggestion, we have added the detection of classical markers of fibrogenesis. Using liquid chromatography-tandem mass spectrometry (LC-MS/MS) for secretome analysis, we found that most of the core ECM proteins (collagens, ECM glycoproteins) are more abundant in cells cultured on stiff substrates (Fig.R1a, b). Those markers were previously identified in CCl₄-treated fibrotic mice liver⁷. We have also evaluated the expression of other fibrogenesis markers (such as *ITGA6*, *LOX*, *FN1*, *COL8A1*, *CTGF*, etc.)²³⁻²⁷.

Fig.R1 (a–b) The differentially expressed proteins in cells on the stiff vs. soft matrix for 2 days, and for 4 days. (The proteins of collagens and ECM glycoproteins shown in this figure were identified in CCl₄-treated fibrotic mice liver⁷.)

The gene *ITGB2* has been reported to be upregulated in systemic sclerosis²⁸ and colorectal fibrosis²⁹, but has not been reported in hepatic fibrosis. In Fig.2c, we showed the ATAC-seq/RNA-seq profile of classical fibrosis markers (*COL8A1* and *ACTG2*)¹⁸⁻¹⁹, and we also showed *ITGB2* gene to propose a possible new finding. Further explanation on the role of *ITGB2* in HSC fibrogenesis may need more evidence. To better align with the main findings of our research and to present a more balanced and convincing data, we have added more well-known fibrotic genes in the revised manuscript (main text Fig.1e–f, Fig.2d, and Fig.3c)

Fig.R2 (a–b) Heatmaps of RNA-seq datasets for cells cultured on different matrices for 2 or 4 days. Marker genes of fibrogenesis are listed on the right. (This figure is the revised version of the main text Fig.1e–f). **(c)** Normalized H3K4me3, H3K27me3, and H3K27ac signal profiles at a locus in *TIMP4* are shown together with a normalized ATAC-seq profile. The vertical yellow boxes highlight the transcription start sites. (This figure is the revised version of the main text Fig.3c)

2. The authors focus on JUN and ERK as key regulators, with good evidence for a role for JUN (in an assumed AP-1 complex) and inhibitor evidence for a role for prolonged activation of ERK. However, this conclusion seems over-simplified because the authors have not shown that simple experimental elevation of p-ERK by some stimulus other than stiffness will elevate p-JUN and lead to chromatin remodeling. The authors seem to acknowledge this weakness, but it appears to be quite significant and weakens their conclusions. At a minimum, if no experimental support can be provided, they would need to emphasize that their findings are descriptive and correlative because inhibitor studies can only identify a requirement for some process, not causality.

Response: We sincerely appreciate the insightful comments regarding our focus on JUN and ERK as key regulators. By using the inhibitor, our work demonstrated that ERK activation is required by the mechano-transduction process, which to some extent illustrates the relationship between ERK and the cell's response to matrix stiffness.

In response to reviewer's comment, we made more discussion on the limitations of our approach by using inhibitor and the need for future studies to explore the causal relationships more comprehensively, such as by conducting experiments using alternative stimuli to manipulate the activation levels of p-ERK and observing the corresponding effects on p-JUN and chromatin remodeling (main text lines 468–471).

Less major:

3. The authors are in a chicken-versus-egg situation in that matrix stiffness itself is the result of transcriptional changes in matrix secretion, so it would help if they would clarify why they are studying only cell responses to stiffness with respect to fibrogenesis. Most likely, they are examining part of a positive feedback cycle of cell activation, matrix synthesis and assembly to increase stiffness, which in turn induces their types of response with more matrix synthesis.

Response: Thanks for reviewer's insightful comment. We clarified the purpose of our study on the cell response to matrix stiffness in the revised manuscript (main text lines 70–78). It is known that the mechanism of liver fibrosis and the occurrence of liver cirrhosis pivot around the activation and differentiation of HSCs into myofibroblasts, along with their simultaneous synthesis and secretion of large amounts of ECM. During

the progression of fibrosis, changes in biomechanical factors are mainly reflected in the reorganization and stiffening of the ECM. The continuous accumulation of ECM eventually results in an increase in tissue stiffness. Crucially, HSCs have the ability to sense the changes in stiffness, which then trigger their activation and prompt them to secrete even more ECM, thereby creating a vicious cycle³⁰.

Our focus on studying cell responses to stiffness in the context of fibrogenesis is precisely because stiffness acts as a key trigger in this vicious cycle. By investigating how cells respond to changes in stiffness, we aim to understanding the initial and crucial steps that set off and drive the fibrogenesis process. It's our starting point to dissect this complex feedback loop that encompasses cell activation, matrix synthesis, and the resulting increase in stiffness. While we acknowledge that this is just one part of the whole picture, we believe it provides essential insights into the fundamental mechanisms underlying fibrogenesis.

4. The ITGB2 marker they use is often a marker for certain immune cells, and it is puzzling that they also identify a major gene ontology category of "antigen processing and presentation" by their ATAC-seq and RNA-seq analyses. How can this strange GO category be explained for an HSC line?

Response: Thanks for reviewer's insightful comment. Our results revealed that the "antigen processing and presentation" GO category is enriched among the upregulated genes in cells cultured on a soft matrix (in other words, these are the genes that are downregulated on a stiff matrix) (main text, Fig.2c). This is because it has been proposed that various hepatic parenchymal and non-parenchymal cell types serve as liver-resident antigen-presenting cells (APCs) for T cells^{31,32}. Furthermore, Martin *et al.* have found that hepatocytes, hepatic stellate cells (HSCs) and liver sinusoidal endothelial cells (LSECs) can present metabolite antigens to T cells *via* major histocompatibility complex-like molecule MR1²⁴. Upon examining the genes within the "antigen processing and presentation" category, the MR1 gene is involved in the list and ranks highly. Therefore, our interpretation is that HSCs can function as APCs during the process of liver fibrosis, which is consistent with previous research findings.

5. Minor typos on line 457: "could response to the stiffen matrix"

Response: Thanks for catching the typos. We have checked throughout the manuscript and made revisions to the corresponding content in the manuscript “could respond to the stiff matrix”.

- 1 Caliari, S. R. et al. Gradually softening hydrogels for modeling hepatic stellate cell behavior during fibrosis regression. *Integr Biol-Uk* **8**, 720-728 (2016).
- 2 Killaars, A. R. et al. Extended Exposure to Stiff Microenvironments Leads to Persistent Chromatin Remodeling in Human Mesenchymal Stem Cells. *Adv Sci (Weinh)* **6**, 1801483 (2019).
- 3 Walker, C. J. et al. Nuclear mechanosensing drives chromatin remodelling in persistently activated fibroblasts. *Nat Biomed Eng* (2021).
- 4 Walker, C. J. et al. Extracellular matrix stiffness controls cardiac valve myofibroblast activation through epigenetic remodeling. *Bioeng Transl Med* **7**, e10394 (2022).
- 5 Kloxin, A. M., Kasko, A. M., Salinas, C. N. & Anseth, K. S. Photodegradable hydrogels for dynamic tuning of physical and chemical properties. *Science* **324**, 59-63 (2009).
- 6 Naba, A. et al. The matrisome: in silico definition and in vivo characterization by proteomics of normal and tumor extracellular matrices. *Mol Cell Proteomics* **11**, M111 014647 (2012).
- 7 Wu, Y. et al. Dynamically remodeled hepatic extracellular matrix predicts prognosis of early-stage cirrhosis. *Cell Death Dis* **12**, 163 (2021).
- 8 Wang C, Z. F., Shan B, Liu J, Zhu L. Real-time measurement of cell contractile force during activation of human hepatic stellate cell line LX-2. *Journal of biomedical engineering* **36**, 841–849 (2019).
- 9 Wang, Y. et al. LIMD1 phase separation contributes to cellular mechanics and durotaxis by regulating focal adhesion dynamics in response to force. *Dev Cell* **56**, 1313-1325 e1317 (2021).
- 10 Ribeiro, M. C. et al. A cardiomyocyte show of force: A fluorescent alpha-actinin reporter line sheds light on human cardiomyocyte contractility substrate stiffness. *J Mol Cell Cardiol* **141**, 54-64 (2020).
- 11 Zhang, M., Dong, X. Y., Wei, Q., Ye, Y. X. & Zhou, H. Hydrogel stiffness mediates the PI3K-AKT signaling of mouse bone marrow stromal cells through cellular traction force. *Colloid Interfac Sci* **62** (2024).
- 12 Jones, D. L. et al. ZNF416 is a pivotal transcriptional regulator of fibroblast mechanoactivation. *J Cell Biol* **220** (2021).
- 13 Jones, D. L. et al. Mechanoepigenetic regulation of extracellular matrix homeostasis via Yap and Taz. *Proc Natl Acad Sci U S A* **120**, e2211947120 (2023).
- 14 Jain, I., Brougham-Cook, A. & Underhill, G. H. Effect of distinct ECM microenvironments on the genome-wide chromatin accessibility and gene expression responses of hepatic stellate cells. *Acta Biomater.* **167**, 278-292 (2023).
- 15 Gerardo, H. et al. Soft culture substrates favor stem-like cellular phenotype and facilitate reprogramming of human mesenchymal stem/stromal cells (hMSCs) through mechanotransduction. *Sci Rep-Uk* **9** (2019).
- 16 Stowers, R. S. et al. Matrix stiffness induces a tumorigenic phenotype in mammary epithelium through changes in chromatin accessibility. *Nature Biomedical Engineering* **3**,

1009-1019 (2019).

- 17 Kordes, C., Sawitza, I., Götze, S., Herebian, D. & Häussinger, D. Hepatic stellate cells contribute to progenitor cells and liver regeneration. *Journal of Clinical Investigation* **124**, 5503-5515 (2014).
- 18 Nakhaei-Rad, S. et al. The Role of Embryonic Stem Cell-expressed RAS (ERAS) in the Maintenance of Quiescent Hepatic Stellate Cells. *J. Biol. Chem.* **291**, 8399-8413 (2016).
- 19 Wernig, G. et al. Unifying mechanism for different fibrotic diseases. *Proceedings of the National Academy of Sciences* **114**, 4757-4762 (2017).
- 20 Schulien, I. et al. The transcription factor c-Jun/AP-1 promotes liver fibrosis during non-alcoholic steatohepatitis by regulating Osteopontin expression. *Cell Death Differ* **26**, 1688-1699 (2019).
- 21 Adler, V., Franklin, C. C. & Kraft, A. S. Phorbol esters stimulate the phosphorylation of c-Jun but not v-Jun: regulation by the N-terminal delta domain. *Proc Natl Acad Sci U S A* **89**, 5341-5345 (1992).
- 22 Ernst, J. & Kellis, M. ChromHMM: automating chromatin-state discovery and characterization. *Nature Methods* **9**, 215-216 (2012).
- 23 Liu, X. Y. et al. Fibronectin expression is critical for liver fibrogenesis and. *Mol Med Rep* **14**, 3669-3675 (2016).
- 24 Ramachandran, P. et al. Resolving the fibrotic niche of human liver cirrhosis at single-cell level. *Nature* **575**, 512-518 (2019).
- 25 Abou-Shady, M. et al. Connective tissue growth factor in human liver cirrhosis. *Liver* **20**, 296-304 (2000).
- 26 Rosenthal, S. B. B. et al. Multi-Modal Analysis of Disease State in Human Hepatic Stellate Cells Identifies Novel Therapeutic Targets. *Hepatology* **76**, S102-S103 (2022).
- 27 El Taghdouini, A. et al. Genome-wide analysis of DNA methylation and gene expression patterns in purified, uncultured human liver cells and activated hepatic stellate cells. *Oncotarget* **6**, 26729-26745 (2015).
- 28 Xu, D., Li, T., Wang, R. & Mu, R. Expression and Pathogenic Analysis of Integrin Family Genes in Systemic Sclerosis. *Front Med (Lausanne)* **8**, 674523 (2021).
- 29 Huang, S. et al. TMT-labelled quantitative proteomic analysis to identify the proteins underlying radiation-induced colorectal fibrosis in rats. *J Proteomics* **223**, 103801 (2020).
- 30 Herum, K. M., Lunde, I. G., McCulloch, A. D. & Christensen, G. The Soft-and Hard-Heartedness of Cardiac Fibroblasts: Mechanotransduction Signaling Pathways in Fibrosis of the Heart. *J Clin Med* **6** (2017).
- 31 Jeffery, H. C. et al. Biliary epithelium and liver B cells exposed to bacteria activate intrahepatic MAIT cells through MR1. *Journal of Hepatology* **64**, 1118-1127 (2016).
- 32 Lett, M. J. et al. Stimulatory MAIT cell antigens reach the circulation and are efficiently metabolised and presented by human liver cells. *Gut* **71**, 2526-2538 (2022).

Point-to-point Response

We appreciate the valuable comments and suggestions from the reviewers. We have used them as a springboard to improve our manuscript by performing new experiments, strengthening the rigor of our statistical analyses, enhancing the lucidity of our data presentation, clarifying ambiguous descriptions of experimental details and analysis procedures, as well as explicating potential sources of confusion in our manuscript.

In response to these comments, we have substantially revised our manuscript, as detailed in our point-by-point response to editor's and reviewers' comments. Please note that all modifications in the manuscript are highlighted with red (the first revision) or yellow (the second revision) for your easy reference.

Reviewers' comments:

Reviewer #1 (Remarks to the Author):

I thank the authors for addressing my concerns and comments. The newly generated data and discussions greatly strengthen the claims in the manuscript.

My last remaining suggestion is to include the additional newly generated data & figures located in the rebuttal letter (eg Figure R4) into the manuscript, either in the main figures or supplement.

Also, the proteomics data of cells on soft vs stiff matrices is powerful data and quite interesting and should go in the main figures.

Response: Thank you very much for your positive feedback and for acknowledging the efforts we have made to address your concerns and comments.

In response to your suggestion, we have accordingly included Figure R4 in the **Supplement Fig.1e–f**, ensuring that it is properly contextualized and discussed to contribute to the overall narrative of our study. Furthermore, we have taken your advice regarding the proteomics data of cells on soft versus stiff matrices, promoting it from the supplementary material to the main **Fig.1g–h**. We believe this will highlight the importance of these findings and provide a more comprehensive view of our research to the readers.

Reviewer #2 (Remarks to the Author):

The authors have responded fully and conscientiously to my own comments. I support acceptance for publication if the other reviewer is also sufficiently positive.

Very minor point:

In lines 468-470, there are two very minor grammatical issues (play vs. plays and probably a period instead of the comma): “We found that the activated p-JUN play a key role in transducing the mechanical stimuli to the nucleus, further studies are needed to explore whether p-ERK nuclear activity sufficient to overcome soft matrix effects”

Response: We would like to express our gratitude for your thorough review and for your support in recommending our manuscript for publication. Your conscientious feedback has been invaluable in refining our work.

Regarding the very minor grammatical issues you pointed out, we have made the necessary corrections. Specifically, we have addressed the issues in lines 468-470 as follows:

“We found that the activated p-JUN plays a key role in transducing the mechanical stimuli to the nucleus. Further studies are needed to explore whether p-ERK nuclear activity sufficient to overcome soft matrix effects, and whether this explicitly requires JUN phosphorylation.”